# Improvement of PMSM Sensorless Control Based on Synergetic and Sliding Mode Controllers Using a Reinforcement Learning Deep Deterministic Policy Gradient Agent †

Marcel Nicola [1] , Claudiu-Ionel Nicola [1,2,*] and Dan Selișteanu [2]

1 Research and Development Department, National Institute for Research, Development and Testing in Electrical Engineering—ICMET Craiova, 200746 Craiova, Romania; marcel_nicola@yahoo.com or marcel_nicola@icmet.ro

2 Department of Automatic Control and Electronics, University of Craiova, 200585 Craiova, Romania; dansel@automation.ucv.ro

* Correspondence: nicolaclaudiu@icmet.ro or claudiu@automation.ucv.ro

† The present work is an extension of the paper: "Improvement of PMSM Control Using Reinforcement Learning Deep Deterministic Policy Gradient Agent". Presented at the Ee 2021, Novi Sad, Serbia, 27–30 October 2021.

**Abstract:** The field-oriented control (FOC) strategy of a permanent magnet synchronous motor (PMSM) in a simplified form is based on PI-type controllers. In addition to their low complexity (an advantage for real-time implementation), these controllers also provide limited performance due to the nonlinear character of the description equations of the PMSM model under the usual conditions of a relatively wide variation in the load torque and the high dynamics of the PMSM speed reference. Moreover, a number of significant improvements in the performance of PMSM control systems, also based on the FOC control strategy, are obtained if the controller of the speed control loop uses sliding mode control (SMC), and if the controllers for the inner control loops of $i_d$ and $i_q$ currents are of the synergetic type. Furthermore, using such a control structure, very good performance of the PMSM control system is also obtained under conditions of parametric uncertainties and significant variations in the combined rotor-load moment of inertia and the load resistance. To improve the performance of the PMSM control system without using controllers having a more complicated mathematical description, the advantages provided by reinforcement learning (RL) for process control can also be used. This technique does not require the exact knowledge of the mathematical model of the controlled system or the type of uncertainties. The improvement in the performance of the PMSM control system based on the FOC-type strategy, both when using simple PI-type controllers or in the case of complex SMC or synergetic-type controllers, is achieved using the RL based on the Deep Deterministic Policy Gradient (DDPG). This improvement is obtained by using the correction signals provided by a trained reinforcement learning agent, which is added to the control signals $u_d$, $u_q$, and $i_{qref}$. A speed observer is also implemented for estimating the PMSM rotor speed. The PMSM control structures are presented using the FOC-type strategy, both in the case of simple PI-type controllers and complex SMC or synergetic-type controllers, and numerical simulations performed in the MATLAB/Simulink environment show the improvements in the performance of the PMSM control system, even under conditions of parametric uncertainties, by using the RL-DDPG.

**Keywords:** permanent magnet synchronous motor; sliding mode control; synergetic control; reinforcement learning; deep neural networks

## 1. Introduction

Interest in the study of the PMSM and its applications has increased in the past decade due to its constructive advantages, such as small size, low harmonic distortion, high torque density, and easy cooling methods. The increased interest in the PMSM is also supported

by the fact that it can be used in the aerospace industry, computer peripherals, numerically controlled machines, robotics, electric drives, etc. [1–7].

Naturally, the study of PMSM control has also attracted, in particular, the attention of researchers involved in the study and development of applications in the above-mentioned fields. Thus, we can mention the FOC-type control structure, where, in the simplest case, a PI-type controller is used for the outer PMSM rotor speed control loop, and hysteresis ON/OFF-type controllers are used for the inner current control loop. Moreover, a number of types of control systems have been implemented for the PMSM control, including: the adaptive control [8–10], the predictive control [11–13], the robust control [14,15], the back-stepping control [16], the SMC [17,18], the synergetic control [19,20], the fuzzy control [21], the neuro-fuzzy control [22], the artificial neural network control (ANN) [23–25], and the control based on particle swarm optimization (PSO) or genetic algorithms [26].

In addition, in terms of the PMSM sensorless control, a series of PMSM rotor speed observers have been developed in order to eliminate hardware systems specific to measuring transducers, thus providing increased system reliability. The types of observers used include Luenberger [27], model reference adaptive system (MRAS) [28], and sliding mode observer (SMO) speed observers [29,30]. Kalman-type observers are used for the systems described by stochastic means [31].

The intelligent control system has a special role in the development of and improvement in control systems. These include RL agents, which are characterized by the fact that they do not use the mathematical description of the controlled system, but use input signals that contain information on the state of the system and provide control actions to optimize a Reward that consists of signals containing information on the controlled process [32–37]. Moreover, building on four papers by the authors based on the RL agent, which demonstrates the capacity of this type of algorithm to improve the performance of the linear control algorithms of the PMSM (the PI type algorithm [26,38], LQR control algorithm [39], and Feedback Linearization control algorithm [40]), this article demonstrates that a cascade control structure containing nonlinear SMC and synergetic control algorithms can also provide improved performance by using an RL agent.

This paper is a follow-up to a paper [38] presented at the 21st International Symposium on Power Electronics (Ee 2021), which shows an improvement in the performance of the FOC-type control structure of the PMSM by using an RL agent. The type of RL used is the Twin-Delayed Deep Deterministic Policy Gradient (TD3) agent, which is an extension and improved variant of the DDPG agent [41]. Moreover, based on the fact that the FOC-type control structure, in which the controller of the outer speed control loop is of the SMC type and the controllers of the inner current control loop are of the synergetic type, provides peak performance of the control system [42,43], this paper also presents the improvement in the performance of this control structure by using an RL agent. The commonly used machine learning (ML) techniques are the following: Linear Regression, Decision Tree, Support Vector Machine, Neural Networks and Deep Learning, and Ensemble of Tree. Generally, the three main types of machine learning strategies are the following: unsupervised learning, supervised learning, and RL. First, unsupervised learning is typically utilized in the fields of data clustering and dimensionality reduction. Supervised learning deals mainly with classification and regression problems. RL is a framework for learning the relationship between states and actions. Ultimately, the agent can maximize the expected Reward and learn how to complete a mission. This is a very strong analogy with the control of an industrial process. Although the other types of ML can be used to estimate certain parameters, RL is recommended for the control of an industrial process. It is clear that RL is an especially powerful form of artificial intelligence, and we are sure to see more progress from these teams. Moreover, the RL-TD3 used in this paper is an improved version of the Deep Deterministic Policy Gradient (DDPG) agent and is considered the most suitable RL agent for industrial process control [41]. In addition, the authors' papers [26,38–40] compare the results of control systems improved by using the RL-TD3 agent or optimized control law parameters by means of computational intelligence algorithms: particle swarm

optimization (PSO) algorithm, simulated annealing (SA) algorithm, genetic algorithm (GA), and gray wolf optimization (GWO) algorithm. The conclusion clearly shows the superiority of the performance of the PMSM control system the in case of using the RL-TD3 agent. However, we consider that the control structures and the comparative results presented are relevant for this paper.

- The main contributions of this paper are as follows:
- The proposal of a PMSM control structure, where the controller of the outer rotor speed control loop is of the SMC type and the controllers for the inner control loops of $i_d$ and $i_q$ currents are of the synergetic type.
- The improvement in the performance of the PMSM control system based on the FOC-type strategy, both when using simple PI-type controllers or in the case of complex SMC or synergetic-type controllers, is achieved using the RL based on the TD3 agent.

The validations of the results are performed through numerical simulations in the MATLAB/Simulink environment to show the improvements in the performance of the PMSM control system even under the conditions of parametric uncertainties, by using the RL-TD3 agent.

The rest of the paper is organized as follows: Section 2 describes the RL for process control, Section 3 presents a correction of the control signals for the PMSM based on the FOC strategy using the RL-TD3 agent, and Section 4 presents a correction of the control signals for the PMSM FOC strategy based on SMC and synergetic control using the RL-TD3 agent. The results of the numerical simulations are presented in Section 5, and Section 6 presents conclusions.

## 2. Reinforcement Learning for Process Control

Generally speaking, RL consists of learning the execution of a task by the host computer if it interacts dynamically with an unknown process. Without explicit programming of the task learning method, the learning process is based on a set of decisions that are made so as to achieve the maximum of the cumulative Reward, whose expression is set a priori. Figure 1 shows the schematic diagram for a RL scenario.

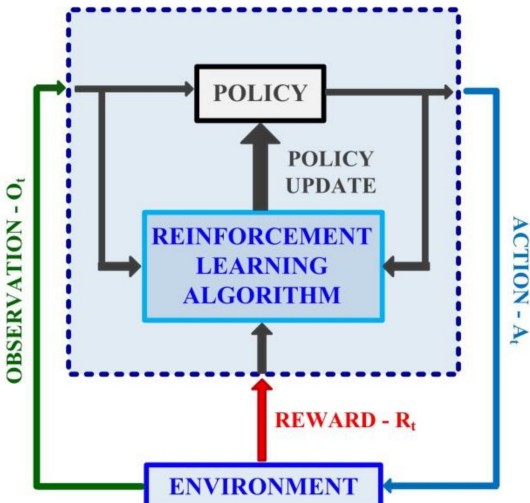

**Figure 1.** Schematic diagram for a RL scenario.

The input signals to the agent are Observation and Reward. The Observation consists of a set of predefined signals characterizing the process, and the Reward is defined as a measure of the success based on the Action output signal. The Action is given by the control quantities of the controlled process. The Observations represents the signals visible for the agent and are found in the form of measured signals, their rate of change, and the related errors.

Usually, the Reward is created as part of the continuous actions in the form of a sum, such as the square of the error of the signals of interest and the square of past actions. These terms are given a weight determined by the final purpose of the problem. Usually, in the process control, the Reward is given by a function that implements the minimization of the steady-state error.

The agent contains a component called Policy, and also implements a learning algorithm. The Policy represents the way in which the actions are associated with the Observations of the process and is described by a function with adjustable parameters. In the case of the process control, the Policy is similar to the operating mode of a controller.

The learning algorithm is designed to find an optimal Policy by continuously updating the parameters of the function associated to the Policy based on the maximization of the cumulative Reward. The process (in the general case referred to as the environment) consists of a plant, reference signals and the associated errors, filters, disturbances, measurement noise, and analog-to-digital and digital-to-analog converters.

The main stages of the RL process are [35–38,41]:

- *Problem statement* defines the learning agent and the possibility of interacting with the process;
- *Process creation* defines the dynamic model of the process and the associated interface;
- *Reward creation* defines the mathematical expression of the Reward in order to measure the performance when undertaken the proposed task;
- *Agent training* is an agent that is trained in order to fulfill the Policy based on the Reward, learning algorithm, and the process.
- *Agent validation* is a step in which the performances are evaluated after the training stage;
- *Deploy policy* is a step that performs the implementation of the trained agent in the control system (for example, generating the executable code for the embedded system).

An agent used in continuous-type systems is the DDPG, whose improved variant, the TD3 agent, is used in this paper. It is characterized by the fact that it is an actor-critic agent [41] that calculates the long-term maximization of the Reward.

In terms of the training stage, the following stages are completed at each step [38,39]:

- For the Observation of the current state $S$, action $A = \mu(S) + N$ is selected, where $N$ is the stochastic noise from the noise model;
- Action $A$ is executed, and Reward $R$ and the next Observation $S'$ are calculated;
- The experience $(S, A, R, S')$ is stored;
- $M$ experiences $(S_i, A_i, R_i, S'_i)$ are randomly generated;
- For $S'_i$, which is a terminal state, the value function target $y_i$ is set to $R_i$.

Otherwise, the following relation is calculated [38,41]:

$$y_i = R_i + \gamma \cdot \min\left( Q'_k\left( S'_k, clip\left(\mu'\left(S'_k|\theta_\mu\right) + \varepsilon\right) \Big| \theta_{Q'_k}\right)\right) \tag{1}$$

The value function target is the sum of the experience Reward $R_i$ and the minimum discounted future Reward from the critics.

At each training step, the parameters of each critic are updated, minimizing the following expression:

$$L_k = \frac{1}{M} \sum_{i=1}^{M} \left(y_i - Q_k\left(S_i, A_i|\theta_{Qk}\right)\right)^2 \tag{2}$$

At each step, the actor's parameters are updated, maximizing the Reward:

$$\nabla_{\theta_\mu} J = \frac{1}{M} \sum_{i=1}^{M} G_{ai} G_{\mu i} \tag{3}$$

where $G_{ai}$, $G_{\mu i}$, and $A$ are given by the following expressions:

$$G_{ai} = \nabla_A \min\left(Q_k\left(S_i, A \middle| \theta_Q\right)\right) \tag{4}$$

$$G_{\mu i} = \nabla_{\theta_\mu} \mu\left(S_i \middle| \theta_\mu\right) \tag{5}$$

$$A = \mu\left(S_i \middle| \theta_\mu\right) \tag{6}$$

Furthermore, the parametric updating is performed, for a chosen smoothing coefficient $\tau$, in the next form:

$$\theta_{Qk'} = \tau\theta_{Qk} + (1 - \tau)\theta_{Qk'} \tag{7}$$

$$\theta_{\mu'} = \tau\theta_\mu + (1 - \tau)\theta_{\mu'} \tag{8}$$

## 3. Correction of the Control Signals for PMSM Based on FOC Strategy Using the RL Agent

The equations of a PMSM are usually expressed using a *d-q* reference frame using the following relations [18]:

$$\begin{cases} \frac{di_d}{dt} = -\frac{R_s}{L_d}i_d + \frac{L_q}{L_d}n_p\omega i_q + \frac{1}{L_d}u_d \\ \frac{di_q}{dt} = -\frac{R_s}{L_q}i_q - \frac{L_d}{L_q}n_p\omega i_d - \frac{\lambda_0}{L_q}n_p\omega + \frac{1}{L_q}u_q \\ \frac{d\omega}{dt} = \frac{3}{2}\frac{n_p}{J}\left(\lambda_0 i_q + (L_d - L_q)i_d i_q\right) - \frac{1}{J}T_L - \frac{B}{J}\omega \\ \frac{d\theta_e}{dt} = n_p\omega \end{cases} \tag{9}$$

where: $u_d$ is the voltage on the *d*-axis, $u_q$ is the voltage the *q*-axis, $R_d$ is the resistance on the *d*-axis, $R_q$ is the resistance on the *q*-axis, $L_d$ is the inductance on the *d*-axis, $L_q$ is the inductance on the *q*-axis, $i_d$ is the current on the *d*-axis, $i_q$ is the current on the *q*-axis, $\omega$ is the PMSM rotor speed, $B$ is the viscous friction coefficient, $J$ is the rotor inertia, $n_p$ is the number of pole pairs, $T_L$ is the load torque, $\lambda_0$ is the flux linkage, and $\theta_e$ is the electrical angle of the PMSM rotor.

The classic control structure of the PMSM is the FOC-type structure, and is presented in Figure 2. In this paper, the RL-TD3 agent is used, which will learn the behavior of the PMSM control system. In turn, this system supplies the correction signals for the three control inputs of the cascade control system ($i_{qref}$, $u_{dref}$, $u_{qref}$) after the training phase, so that the improved control system will have superior performance.

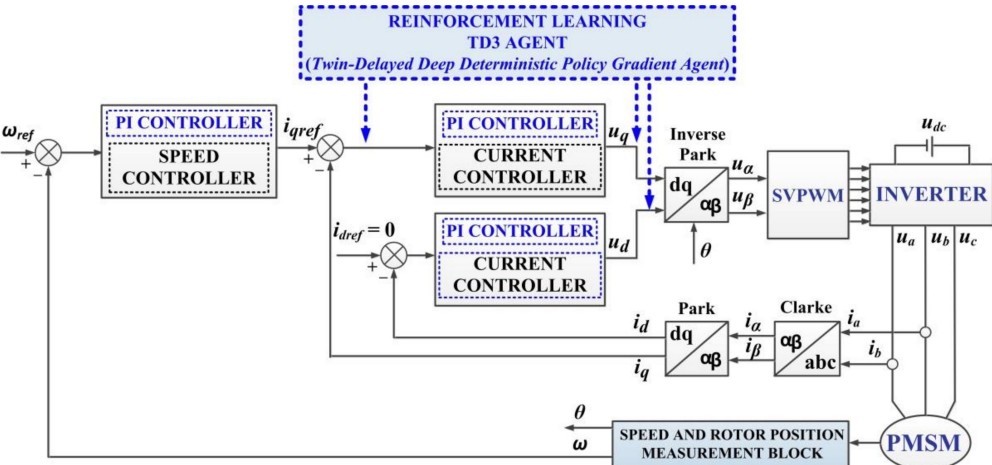

**Figure 2.** Block diagram for FOC-type control of the PMSM based on PI-type controllers using RL.

In the first step, the deep neural network is created, which is characterized by two inputs (Observation and Action) and one output. The code sequence of the program

developed in the MATLAB environment for the creation of the neural network is presented in Figure 3, and its graphic representation is shown in Figure 4.

```
train_Agent.m  ×  +
1    rng(0)
2    statePath = [imageInputLayer([numObservations,1,1],'Normalization','none','Name','State')
3        fullyConnectedLayer(64,'Name','fc1')];
4    actionPath = [imageInputLayer([numActions,1,1],'Normalization', 'none', 'Name','Action')
5        fullyConnectedLayer(64, 'Name','fc2')];
6    commonPath = [additionLayer(2,'Name','add')
7        reluLayer('Name','relu2')
8        fullyConnectedLayer(32, 'Name','fc3')
9        reluLayer('Name','relu3')
10       fullyConnectedLayer(16, 'Name','fc4')
11       fullyConnectedLayer(1, 'Name','CriticOutput')];
12   criticNetwork = layerGraph();
13   criticNetwork = addLayers(criticNetwork,statePath);
14   criticNetwork = addLayers(criticNetwork,actionPath);
15   criticNetwork = addLayers(criticNetwork,commonPath);
16   criticNetwork = connectLayers(criticNetwork,'fc1','add/in1');
17   criticNetwork = connectLayers(criticNetwork,'fc2','add/in2');
18   criticOptions = rlRepresentationOptions('LearnRate',1e-4,'GradientThreshold',1);
19   critic1 = rlQValueRepresentation(criticNetwork,observationInfo,actionInfo,...
20       'Observation',{'State'},'Action',{'Action'},criticOptions);
21   critic2 = rlQValueRepresentation(criticNetwork,observationInfo,actionInfo,...
22       'Observation',{'State'},'Action',{'Action'},criticOptions);
23   actorNetwork = [imageInputLayer([numObservations,1,1],'Normalization','none','Name','State')
24       fullyConnectedLayer(64, 'Name','actorFC1')
25       reluLayer('Name','relu1')
26       fullyConnectedLayer(32, 'Name','actorFC2')
27       reluLayer('Name','relu2')
28       fullyConnectedLayer(numActions,'Name','Action')
29       tanhLayer('Name','tanh1')];
30   actorOptions = rlRepresentationOptions('LearnRate',1e-3,'GradientThreshold',1,'L2RegularizationFactor',0.001);
31   actor = rlDeterministicActorRepresentation(actorNetwork,observationInfo,actionInfo,...
32       'Observation',{'State'},'Action',{'tanh1'},actorOptions);

Command Window
```

**Figure 3.** Creation of the deep neural network—MATLAB syntax code example.

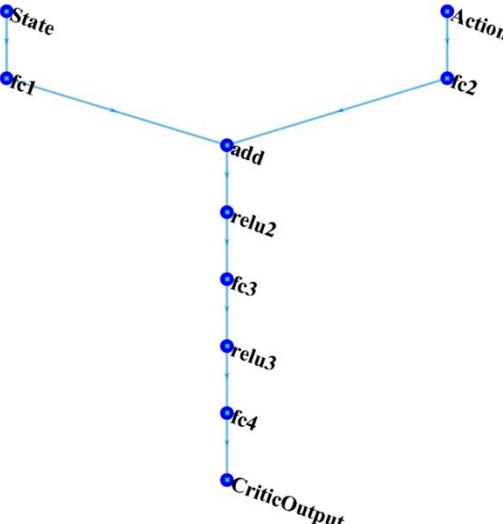

**Figure 4.** Representation of the deep neural network for the RL-TD3 agent.

Training of the TD3 agent for the PMSM control uses a maximum of 200 episodes, the number of steps per episode is 100, and the sampling step of the agent is $10^{-4}$ s. The agent training stage stops when the cumulative average Reward is greater than $-190$ for a period of 100 consecutive episodes, or after the 200 training episodes initially set have passed. To improve the learning performance during training, a Gaussian noise overlaps the signals received and transmitted by the agent.

### 3.1. Reinforcement Learning Agent for the Correction of the Outer Speed Control Loop

The block diagram of the implementation in MATLAB/Simulink for the PMSM control of the outer loop (which controls the speed of PMSM) based on the RL-TD3 agent is presented in Figure 5. Figure 6 shows the RL block structure. In this case, the correction

signals of TD3 agent will be added to the control signal $i_{qref}$. The Observation consists of signals $\omega$ and $\omega_{error}$.

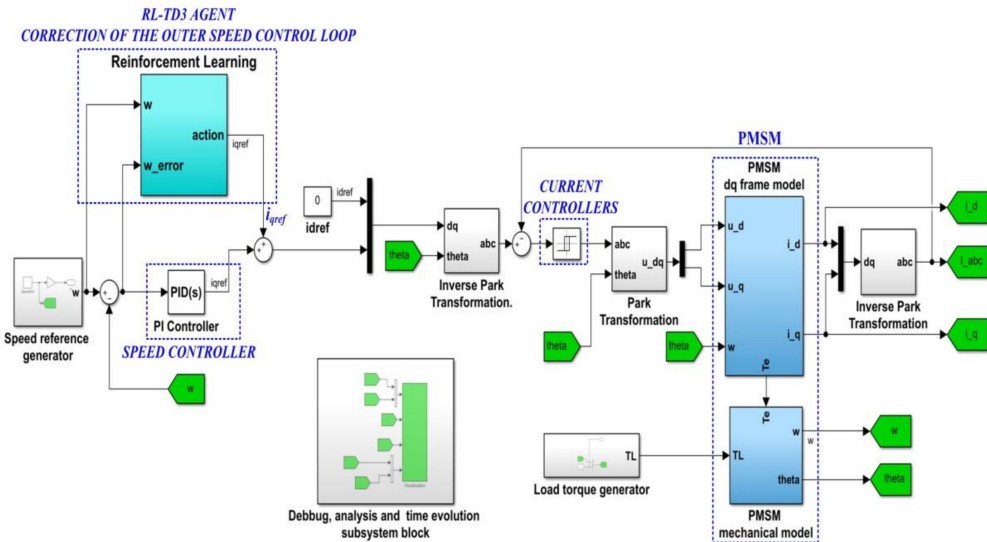

**Figure 5.** Block diagram of the MATLAB/Simulink implementation for the PMSM control based on PI-type controllers using the RL-TD3 agent for the correction of $i_{qref}$.

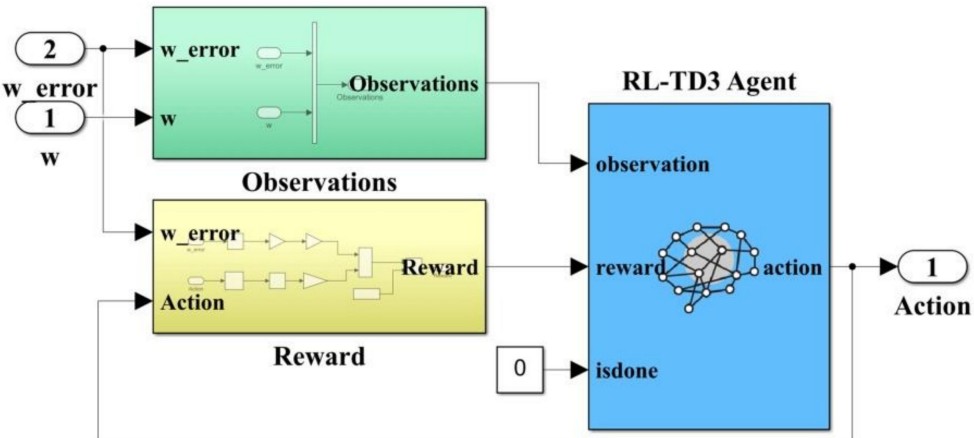

**Figure 6.** MATLAB/Simulink implementation of RL-TD3 agent for the correction of $i_{qref}$.

The Reward at each step for this case is expressed as follows:

$$r_1 = -\left( Q_1 \omega_{error}^2 + R \sum_j \left( u_{t-1}^j \right)^2 \right) \tag{10}$$

where: $Q_1 = 0.5$ and $R = 0.1$.

The training time for this case is 6 h, 23 min, and 27 s. The graphical results for the training stage of the RL-TD3 agent for the correction of $i_{qref}$ are shown in Figure 7.

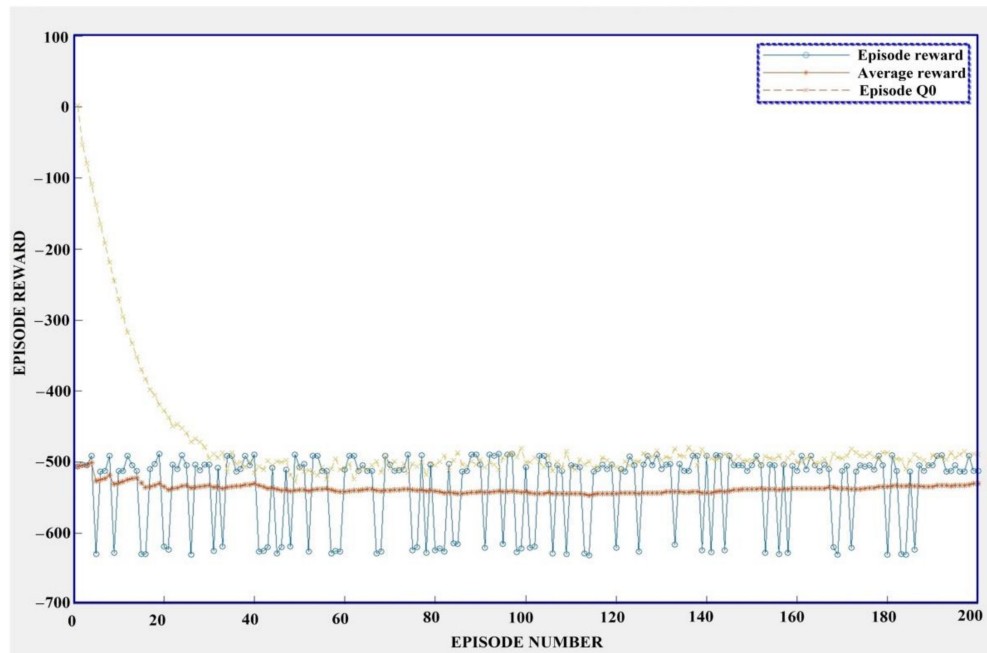

**Figure 7.** Training stage of the RL-TD3 agent for the correction of $i_{qref}$.

### 3.2. Reinforcement Learning Agent for the Correction of the Inner Currents Control Loop

The block diagram of the implementation in MATLAB/Simulink for the PMSM control of the inner loop (which controls the $i_d$ and $i_q$ currents) based on the RL-TD3 agent is presented in Figure 8. After the learning stage, the RL-TD3 agent will supply correction signals for the control signals $u_d$ and $u_q$. Figure 9 shows the RL block structure. The Observation consists of signals $i_d$, $i_q$, $i_{derror}$, and $i_{qerror}$.

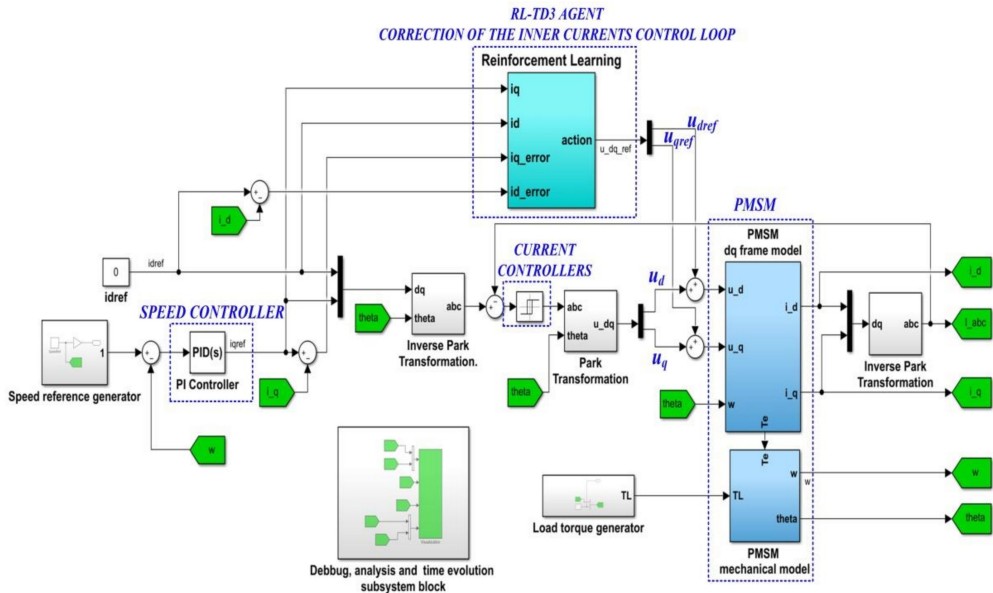

**Figure 8.** Block diagram of the MATLAB/Simulink implementation for the PMSM control based on PI-type controllers using the RL-TD3 agent for the correction of $u_{dref}$ and $u_{qref}$.

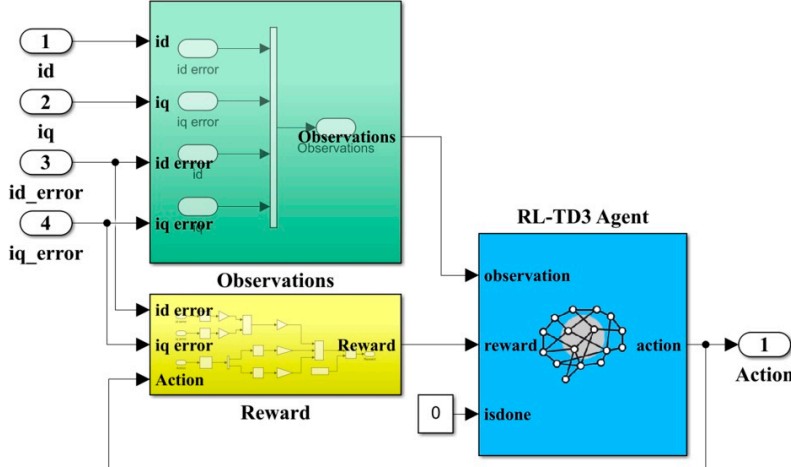

**Figure 9.** MATLAB/Simulink implementation of the RL-TD3 agent for the correction of $u_{dref}$ and $u_{qref}$.

The Reward at each step for this case is expressed as follows:

$$r_1 = -\left( Q_1 i_{derror}^2 + Q_2 i_{qerror}^2 + R\sum_j \left( u_{t-1}^j \right)^2 \right) \tag{11}$$

where: $Q_1 = Q_2 = 0.5$, $R = 0.1$, and $u_{t-1}^j$ represents the actions from the previous time step.

The training time for this case is 6 h, 8 min, and 2 s. The graphical results for the training stage of the RL-TD3 agent for the correction of $u_{dref}$ and $u_{qref}$ are shown in Figure 10.

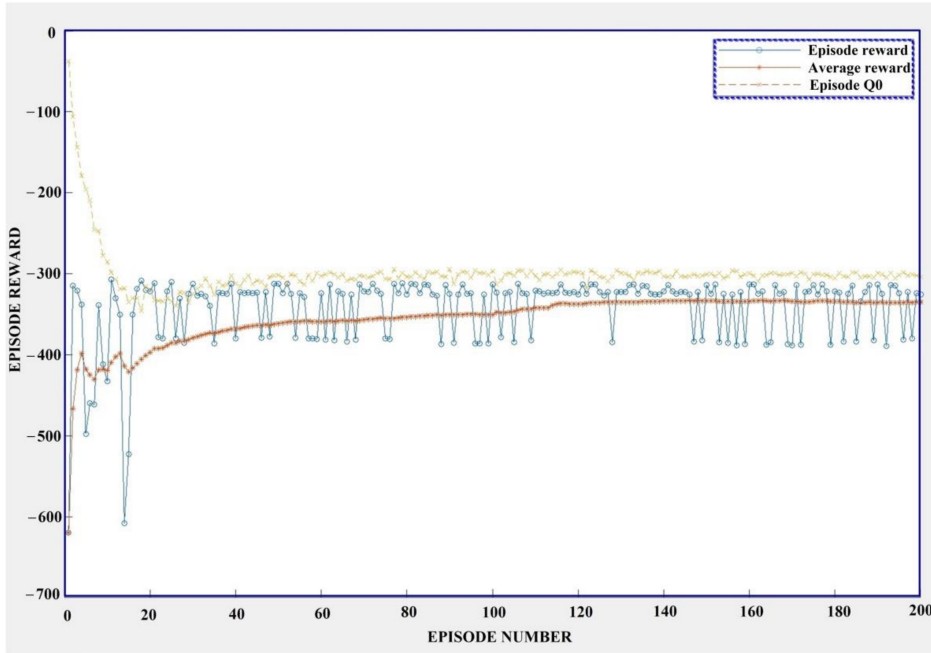

**Figure 10.** Training stage of the RL-TD3 agent for the correction of $u_{dref}$ and $u_{qref}$.

### 3.3. Reinforcement Learning Agent for the Correction of the Outer Speed Control Loop and Inner Currents Control Loop

The block diagram of the implementation in MATLAB/Simulink for the correction of the inner current control loop and outer speed control loop based on the RL-TD3 agent is presented in Figure 11. Figure 12 shows the RL block structure. In this case, the correction

signals of RL-TD3 agent will be supplied to the control signals $u_d$ and $u_q$, and also to the $i_{qref}$ signal. The Observation consists of signals $\omega$, $\omega_{error}$, $i_d$, $i_q$, $i_{derror}$, and $i_{qerror}$.

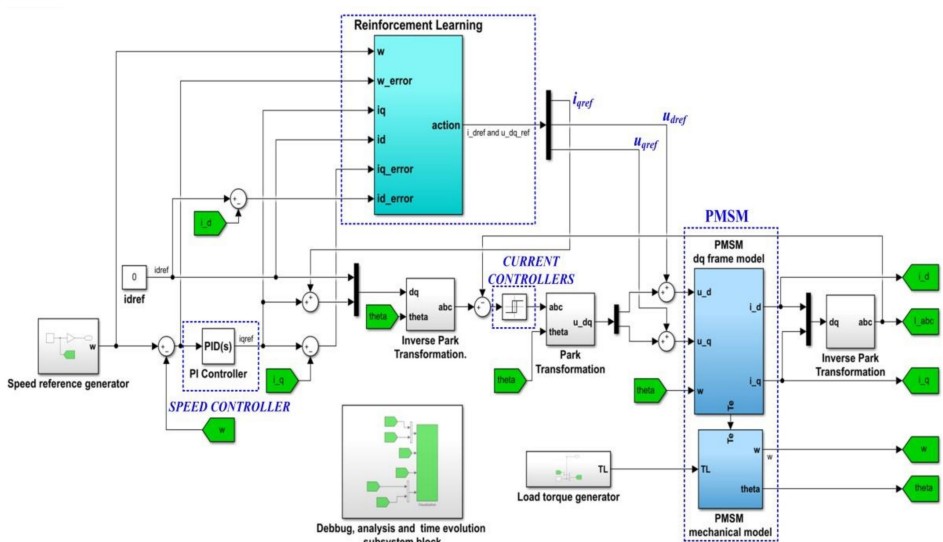

**Figure 11.** Block diagram of MATLAB/Simulink implementation for the PMSM control based on PI-type controllers using the RL-TD3 agent for the correction of $u_{dref}$, $u_{qref}$, and $i_{qref}$.

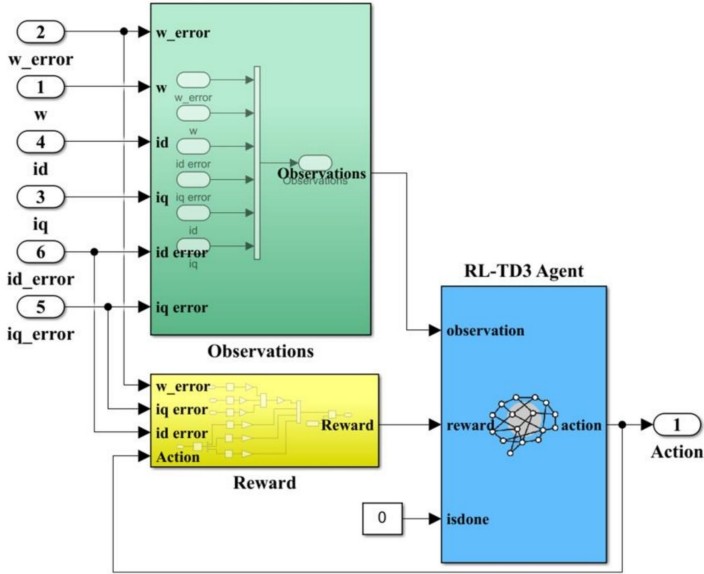

**Figure 12.** MATLAB/Simulink implementation of the RL- TD3 agent for the correction of $u_{dref}$, $u_{qref}$, and $i_{qref}$.

The Reward at each step for this case is expressed as follows:

$$r_1 = -\left( Q_1 \omega_{error}^2 + Q_2 i_{derror}^2 + Q_3 i_{qerror}^2 + R\sum_j \left( u_{t-1}^j \right)^2 \right) \tag{12}$$

where: $Q_1 = Q_2 = Q_3 = 0.5$ and $R = 0.1$.

The training time for this case is 7 h, 59 min, and 35 s. The graphical results for the training stage of the RL-TD3 agent for the correction of $u_d$, $u_q$, $i_{qref}$ are shown in Figure 13.

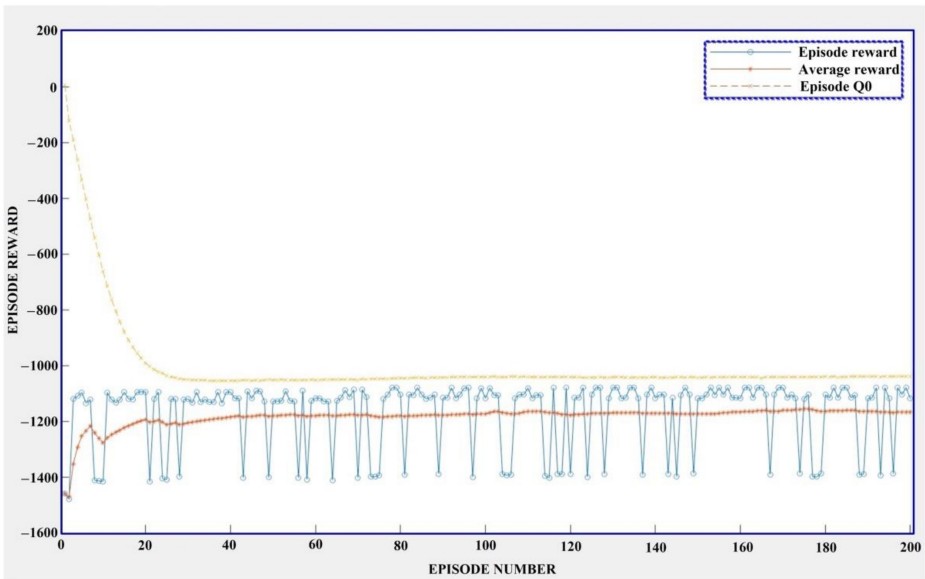

**Figure 13.** Training stage of the RL-TD3 agent for the correction of $u_{dref}$, $u_{qref}$, and $i_{qref}$.

## 4. Correction of the Control Signals for PMSM—FOC Strategy Based on SMC and Synergetic Control Using the Reinforcement Learning Agent

Based on the classic FOC-type structure in which the usual PI or hysteresis controllers are replaced with SMC or synergetic controllers, respectively, Figure 14 presents the block diagram for the SMC and synergetic control of the PMSM based on the reinforcement learning. The following subsection presents the PMSM control in which an SMC-type controller is proposed for the outer speed control loop, and a synergetic-type controller is proposed for the inner current control loop.

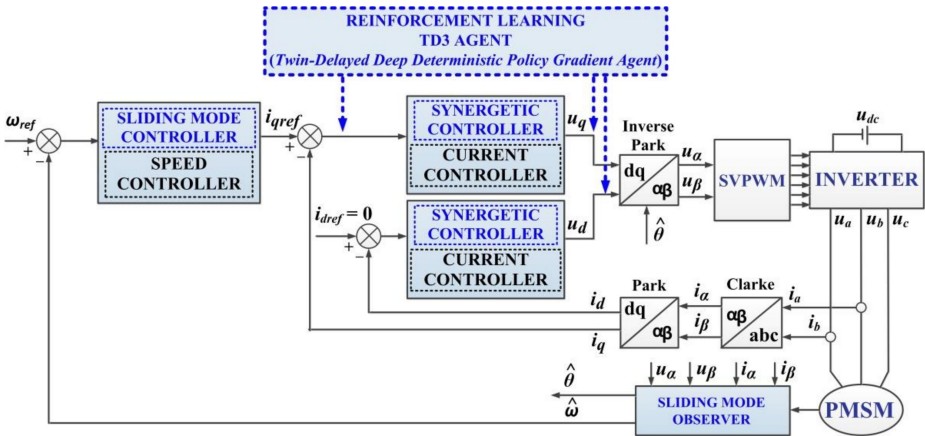

**Figure 14.** Block diagram for the SMC and synergetic control of PMSM based on RL.

### 4.1. SMC and Synergetic for PMSM Control

4.1.1. SMC Speed Controller Description and MATLAB/Simulink Implementation

Based on Equation (9), rewritten in the form of Equation (13), to obtain the SMC control law, the notations are expressed in Equations (14) and (15) in which the state variables $x_1$ and $x_2$ are defined [17,18,42,43].

$$\begin{pmatrix} \dot{i}_d \\ \dot{i}_q \\ \dot{\omega} \end{pmatrix} = \begin{pmatrix} -\frac{R_s}{L} & n_p\omega & 0 \\ -n_p\omega & -\frac{R_s}{L} & -\frac{n_p\lambda_0}{L} \\ 0 & \frac{K_t}{J} & -\frac{B}{J} \end{pmatrix} \begin{pmatrix} i_d \\ i_q \\ \omega \end{pmatrix} + \begin{pmatrix} \frac{u_d}{L} \\ \frac{u_q}{L} \\ -\frac{T_L}{J} \end{pmatrix} \tag{13}$$

$$x_1 = \omega_{ref} - \omega \tag{14}$$

$$x_2 = \dot{x}_1 = \frac{\omega_{ref} - \omega}{dt} = -\dot{\omega} \tag{15}$$

It can be noted that variable $x_1$ represents the tracking error of the PMSM speed and $x_2$ represents its derivative.

The sliding surface $S$ and its derivative are defined in Equations (16) and (17), respectively.

$$S = cx_1 + x_2 \tag{16}$$

$$\dot{S} = cx_2 + \dot{x}_2 = cx_2 - D\dot{i}_q \tag{17}$$

where: $c$ is a positive adjustable parameter and $D = \frac{3}{2}\frac{n_p\lambda_0}{J}$.

In Equation (18), the condition of the evolution on surface $S$ is imposed as follows:

$$\dot{S} = -\varepsilon \text{sgn}(S) - qS, \quad \varepsilon, q > 0 \tag{18}$$

where: $\varepsilon$ and $q$ represent the positive adjustable parameters; *sgn()* represents the *signum* function.

By using the function defined in Equation (19), a reduction in the chattering effect is obtained instead of the function *sgn* [43]:

$$H(x) = \frac{2}{1 + e^{-a(x-c)}} - 1 \tag{19}$$

For $a = 4$ and $c = 0$, $H \in [-1 \ 1]$, the evolution of the function $H$ from $-1$ to $1$ is smoothed, thus reducing the chattering effect.

After some calculations, the SMC-type controller output can be expressed as follows:

$$i_{qref}(t) = \frac{1}{D}\int_0^t [cx_2 + \varepsilon H(S) + qS]dt \tag{20}$$

To demonstrate the stability, a Lyapunov function is chosen of the form $V = \frac{1}{2}S^2$. By calculating the derivative of function $V$, we obtain its negativity according to Equation (21):

$$\dot{V} = S\dot{S} = S[-\varepsilon H(S) - qS] = -\varepsilon H(S) - qS^2 \tag{21}$$

The block diagram of the MATLAB/Simulink implementation for the SMC law control is presented in Figure 15.

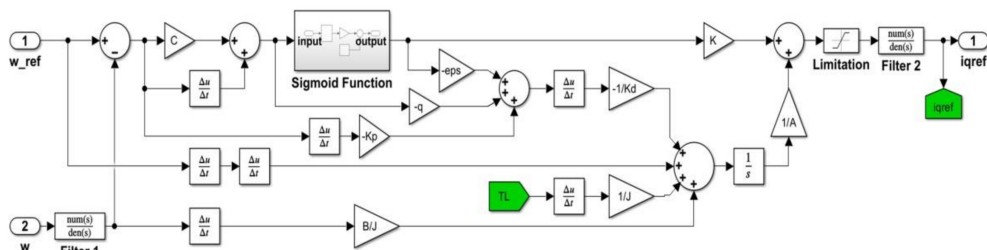

**Figure 15.** Block diagram of the MATLAB/Simulink implementation for the SMC speed controller.

4.1.2. Synergetic Currents Controller Description and MATLAB/Simulink Implementation

By adding a macro-variable as a function of the states of the system in Equation (22) and by imposing an evolution of its dynamics in the form of Equation (23), it is noted that the synergetic control law design is a generalization of SMC control law design [19,42,43]:

$$\Psi = \Psi(x, t) \tag{22}$$

$$T\dot{\Psi} + \Psi = 0, \ T > 0 \tag{23}$$

where: $T$ represents the rate of convergence of the system evolution to achieve the desired manifold.

The differentiation of the macro-variable $\Psi$ can be written by Equation (24):

$$\dot{\Psi} = \frac{d\Psi}{dx}\dot{x}, \tag{24}$$

The general form of the control law for the PMSM is obtained by combining Equations (13), (23) and (24):

$$u = u(x, \Psi(x,t), T, t) \tag{25}$$

Following [20,42], superior performances can be obtained for static and dynamic regimes by selecting threshold values according to Equations (26) and (27):

$$\omega_{acc} = \omega_{ref} - k_q\left(\left|i_{q\max}\right| - i_{qref}\right) \tag{26}$$

$$\omega_{dec} = \omega_{ref} - k_q\left(-\left|i_{q\max}\right| - i_{qref}\right) \tag{27}$$

Based on these threshold values that practically limit the acceleration and deceleration regimes, the macro-variable can be defined on the $q$-axis $\Psi_q$. Therefore, for $\omega > \omega_{acc}$ and $\omega < \omega_{dec}$ the following relation is defined:

$$\Psi_q = \left(\omega(t) - \omega_{ref}\right) + k_q\left(i_q(t) - i_{qref}\right) \tag{28}$$

For $\omega \leq \omega_{acc}$, the corresponding macro-variable is defined as in Equation (29), and for $\omega \geq \omega_{dec}$ the corresponding macro-variable is defined as in Equation (30).

$$\Psi_q = i_q(t) - \left|i_{q\max}\right| + k_{iq}\int_0^t \left(i_q(t) - \left|i_{q\max}\right|\right)dt \tag{29}$$

$$\Psi_q = i_q(t) + \left|i_{q\max}\right| + k_{iq}\int_0^t \left(i_q(t) + \left|i_{q\max}\right|\right)dt \tag{30}$$

where: $i_{q\max}$ represents the maximum admissible current on the $q$-axis, and $k_q$ represents a value that is dynamically adjusted as a function of the PMSM rotor speed error.

Based on these equations, the control law for the $q$-axis can be expressed according to the acceleration and deceleration subdomains presented above, as follows: Equation (31) for $\omega_{acc} < \omega < \omega_{dec}$, $u_q$; Equation (33) for $\omega \leq \omega_{acc}$, $u_q$; and Equation (32) for $\omega \geq \omega_{dec}$, $u_q$:

$$u_q(t) = R_s i_q + n_p\omega(Li_d + \lambda_0) + \frac{L}{T_q}\left(i_{qref} - i_q\right) + \frac{L}{T_q k_q}\left(\omega_{ref} - \omega\right) + \frac{L}{Jk_q}\left(-K_t i_q + B\omega + T_L\right) \tag{31}$$

$$u_q(t) = R_s i_q + n_p\omega(Li_d + \lambda_0) + \frac{L}{T_q}\left(\left|i_{q\max}\right| - i_q\right) + k_{iq}L\left(\left|i_{q\max}\right| - i_q\right) - \frac{k_{iq}L}{T_q}\int_0^t \left(i_q - \left|i_{q\max}\right|\right)dt \tag{32}$$

$$u_q(t) = R_s i_q + n_p\omega(Li_d + \lambda_0) + \frac{L}{T_q}\left(-\left|i_{q\max}\right| - i_q\right) + k_{iq}L\left(-\left|i_{q\max}\right| - i_q\right) - \frac{k_{iq}L}{T_q}\int_0^t \left(i_q + \left|i_{q\max}\right|\right)dt \tag{33}$$

The block diagram of the MATLAB/Simulink implementation for the control law on the $q$-axis corresponding to Equations (31)–(33) is presented in Figures 16–18, respectively.

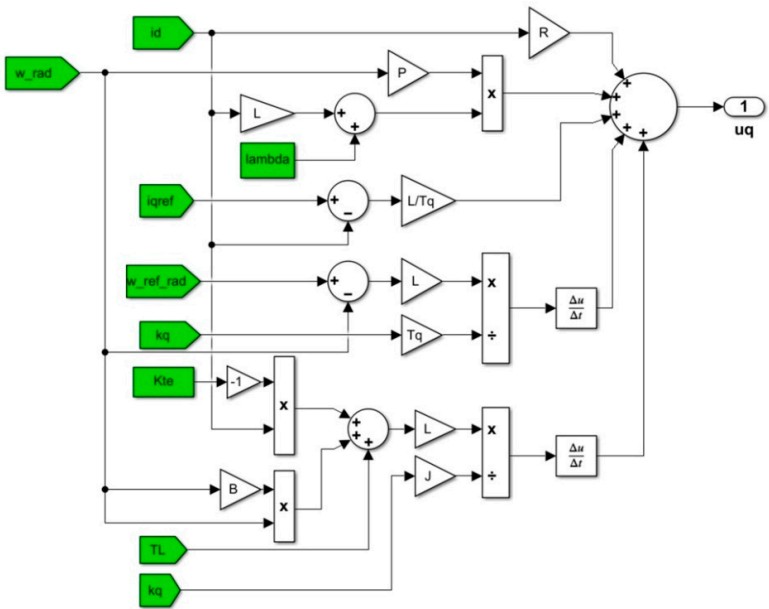

**Figure 16.** Block diagram of the MATLAB/Simulink implementation for the synergetic current controller of the $u_q$ command—Equation (31).

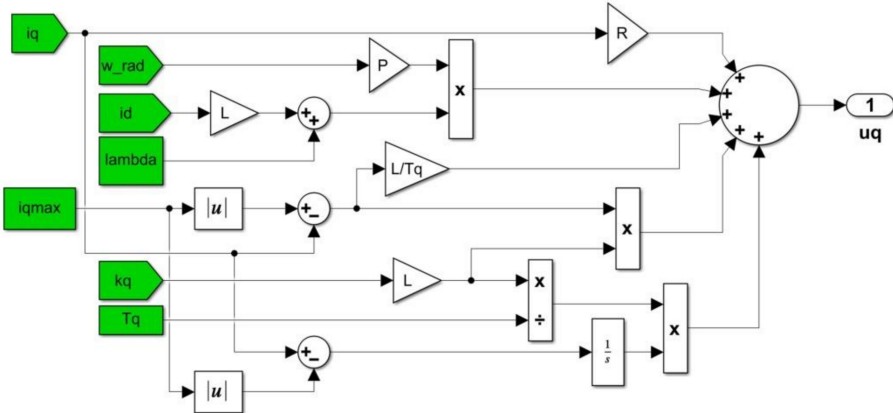

**Figure 17.** Block diagram of the MATLAB/Simulink implementation for the synergetic current controller of the $u_q$ command—Equation (32).

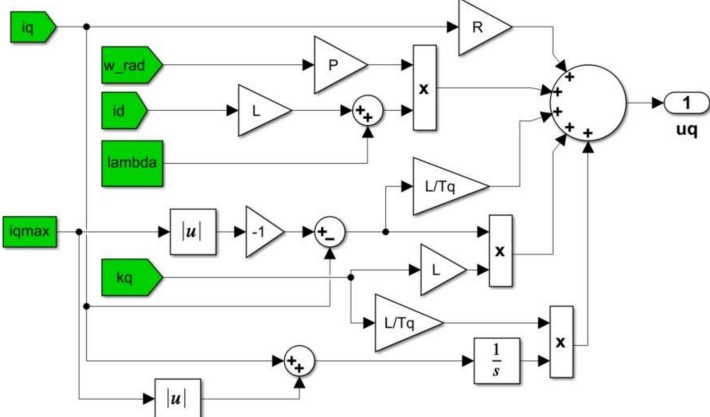

**Figure 18.** Block diagram of the MATLAB/Simulink implementation for the synergetic current controller of the $u_q$ command—Equation (33).

The macro-variable on the $d$-axis $\Psi_d$ can be expressed as in Equation (34):

$$\Psi_d = \left(i_d(t) - i_{dref}\right) + k_{id}\int_0^t \left(i_d(t) - i_{dref}\right)dt \tag{34}$$

After some calculus, the control law for the $d$-axis can be expressed as follows [20,43]:

$$u_d(t) = R_s i_d - n_p \omega L i_q - \frac{L}{T_d}\left(i_d - i_{dref}\right) - k_{id}L\left(i_d - i_{dref}\right) - \frac{k_{id}L}{T_d}\int_0^t \left(i_d - i_{dref}\right)dt \tag{35}$$

The block diagram of the MATLAB/Simulink implementation for the control law on the $d$-axis corresponding to Equation (35) is presented in Figure 19.

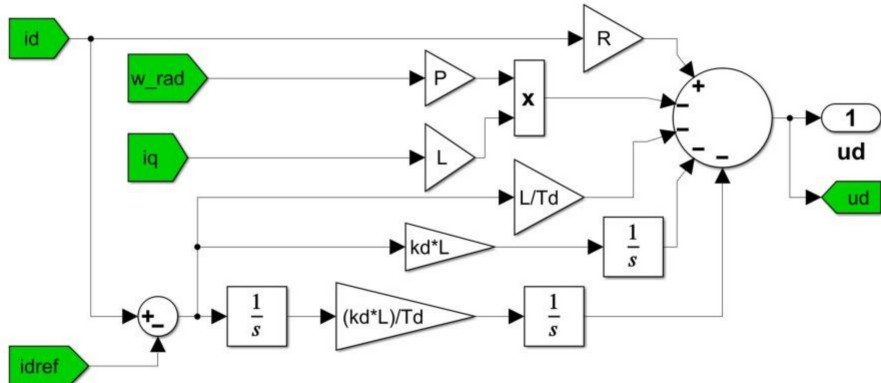

**Figure 19.** Block diagram of the MATLAB/Simulink implementation for the synergetic current controller of the $u_d$ command—Equation (35).

### 4.1.3. Speed Observer

In the $\alpha$-$\beta$ frame, using the PMSM equations derived from Equation (13) after applying the inverse Park transform, Equations (36)–(38) present the expressions of currents $i_\alpha$ and $i_\beta$, their derivatives, and the back-EMF $e_\alpha$ and $e_\beta$ [18,25,43]:

$$\begin{aligned} i_\alpha &= i_d \cos(\theta_e) - i_q \sin(\theta_e) \\ i_\beta &= i_d \sin(\theta_e) + i_q \cos(\theta_e) \end{aligned} \tag{36}$$

$$\begin{aligned} e_\alpha &= \frac{d\lambda_\alpha}{dt} = -\lambda_0 \omega_e \sin(\theta_e) \\ e_\beta &= \frac{d\lambda_\beta}{dt} = -\lambda_0 \omega_e \cos(\theta_e) \end{aligned} \tag{37}$$

$$\begin{aligned} \frac{di_\alpha}{dt} &= -\frac{R_s}{L}i_\alpha - \frac{1}{L}e_\alpha + \frac{1}{L}u_\alpha \\ \frac{di_\beta}{dt} &= -\frac{R_s}{L}i_\beta - \frac{1}{L}e_\beta + \frac{1}{L}u_\beta \end{aligned} \tag{38}$$

The equations of the SMO-type observer can be written as follows [18,25,43]:

$$\begin{aligned} \frac{d\hat{i}_\alpha}{dt} &= -\frac{R_s}{L}\hat{i}_\alpha + \frac{1}{L}u_\alpha - \frac{1}{L}kH(\hat{i}_\alpha - i_\alpha) \\ \frac{d\hat{i}_\beta}{dt} &= -\frac{R_s}{L}\hat{i}_\beta + \frac{1}{L}u_\beta - \frac{1}{L}kH(\hat{i}_\beta - i_\beta) \end{aligned} \tag{39}$$

where: $k$ is the observer gain, and $H$ represents the *sigmoid* type function described in Equation (19).

The sliding vector is chosen as follows:

$$S_n = [S_\alpha\ S_\beta]^T = [\hat{i}_\alpha - i_\alpha\ \hat{i}_\beta - i_\beta]^T = [\bar{i}_\alpha\ \bar{i}_\beta]^T \tag{40}$$

Moreover, a Lyapunov function is chosen in the form of Equation (41) [25,43].

$$V = \frac{1}{2}S_n^T S_n = \frac{1}{2}\left(S_\alpha^2 + S_\beta^2\right) \tag{41}$$

For the calculation of the derivative of this function, the current error system is defined as follows:

$$\begin{aligned}
\dot{\bar{i}}_\alpha &= \dot{\hat{i}}_\alpha - \dot{i}_\alpha = -\frac{R_s}{L}\bar{i}_\alpha + \frac{1}{L}e_\alpha - \frac{1}{L}kH(\bar{i}_\alpha) \\
\dot{\bar{i}}_\beta &= \dot{\hat{i}}_\beta - \dot{i}_\beta = -\frac{R_s}{L}\bar{i}_\beta + \frac{1}{L}e_\beta - \frac{1}{L}kH(\bar{i}_\beta)
\end{aligned} \tag{42}$$

Based on these, Equation (43) is obtained.

$$\dot{V} = -\frac{R_s}{L}\left(\bar{i}_\alpha^2 + \bar{i}_\beta^2\right) + \frac{1}{L}\left[(e_\alpha - k)\bar{i}_\alpha H(\bar{i}_\alpha) + (e_\beta - k)\bar{i}_\beta H(\bar{i}_\beta)\right] < 0 \tag{43}$$

By choosing the observer gain of the form $k \geq \max(|e_\alpha|, |e_\beta|)$, the stability condition of the speed observer is obtained: $\dot{V} < 0$.

In addition, the following relation can be written:

$$[\dot{S}_\alpha \ \dot{S}_\beta]^T = [S_\alpha \ S_\beta]^T \approx [0 \ 0] \tag{44}$$

Using Equations (43) and (44) the estimations for $e_\alpha$ and $e_\beta$ are achieved:

$$\begin{aligned}
\hat{e}_\alpha &= kH(\bar{i}_\alpha) = -\lambda_0\hat{\omega}_e \sin\theta_e \\
\hat{e}_\beta &= kH(\bar{i}_\beta) = \lambda_0\hat{\omega}_e \cos\theta_e
\end{aligned} \tag{45}$$

Equations (46) and (47) show the estimations of the PMSM rotor speed and positions.

$$\hat{\omega}_e = \frac{\sqrt{\hat{e}_\alpha^2 + \hat{e}_\beta^2}}{\lambda_0} \tag{46}$$

$$\hat{\theta}_e(t) = \int_{t_0}^{t} \hat{\omega}_e(t)dt + \theta_0 \tag{47}$$

where: $\theta_0$ is the initial electrical position of the PMSM rotor.

The block diagram of the MATLAB/Simulink implementation for the PMSM rotor speed and position estimations based on the SMO-type observer is presented in Figure 20.

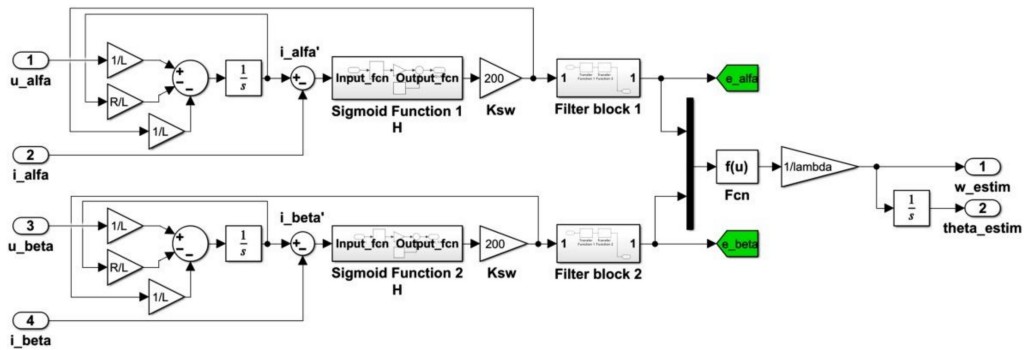

**Figure 20.** Block diagram of the MATLAB/Simulink implementation for PMSM rotor speed and position estimations.

The block diagram of the MATLAB/Simulink implementation for the SMC and synergetic sensorless control of the PMSM (developed based on the elements presented in this subsection) is presented in Figure 21. For the implementation of the SMC-type controller

described in Section 4.1.1, the parameters $\varepsilon = 300$, $q = 200$, and $c = 100$ were selected, and for the synergetic type controller described in Section 4.1.2, the parameters $k_{iq} = 10{,}000$, $k_q = 10{,}000$, $i_{qmax} = 50$, $T_d = 3$, $T_q = 3$, and $k_{id} = 10{,}000$ were selected.

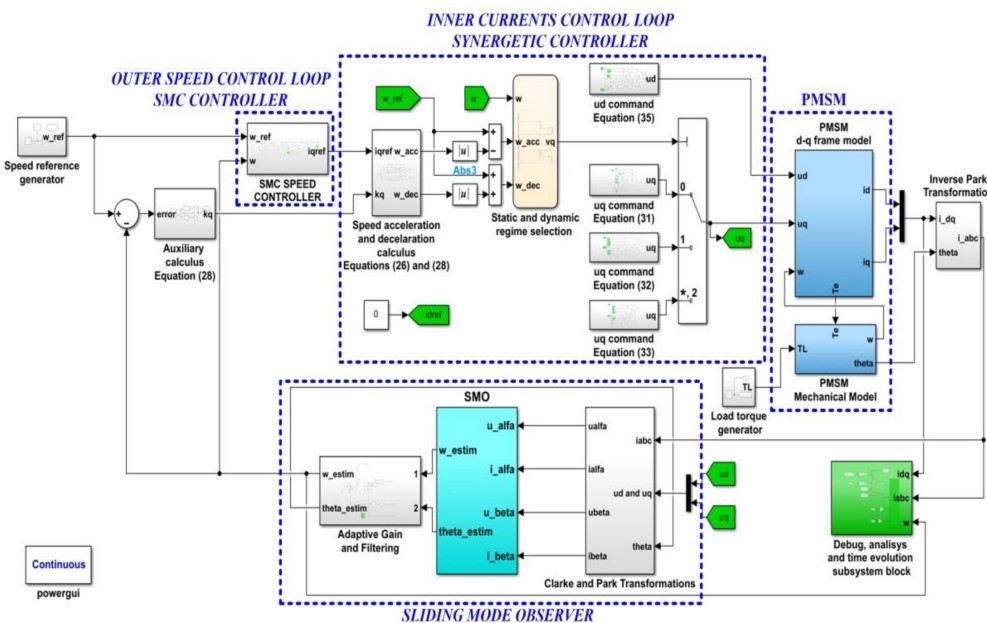

**Figure 21.** Block diagram of the MATLAB/Simulink implementation for SMC and synergetic sensorless control of the PMSM.

## 4.2. Reinforcement Learning Agent for the Correction of the Outer Speed Control Loop Using SMC and Synergetic Control

By analogy with the aspects presented in Section 3, this section continues with the presentation of the way in which the performance of the PMSM control system can be improved using the RL, even if the controllers used in the FOC-type control strategy are complex SMC and synergetic controllers. A maximum number of 200 training episodes was chosen for the RL-TD3 agent, and for each episode the number of steps is 100, with a sampling period of $10^{-4}$ s. The RL-TD3 agent training stage is stopped when the cumulative average Reward is greater than $-190$ for a period of 100 consecutive episodes, or after 200 training episodes have run. A Gaussian noise is overlapped on the original signals, received and transmitted by the agent, to improve the learning performance.

The block diagram of the implementation in MATLAB/Simulink for the PMSM control using SMC and synergetic controllers, resulting in improved performance using the RL-TD3 agent in the outer control loop, is presented in Figure 22.

The correction signals of the RL-TD3 agent will be added to the control signal $i_{qref}$, the RL block structure is similar to that of Figure 6, and the Reward is given by Equation (10). The training time for this case is 10 h, 17 min, and 31 s. The graphical results for the training stage of RL-TD3 agent for the correction of $i_{qref}$ are shown in Figure 23.

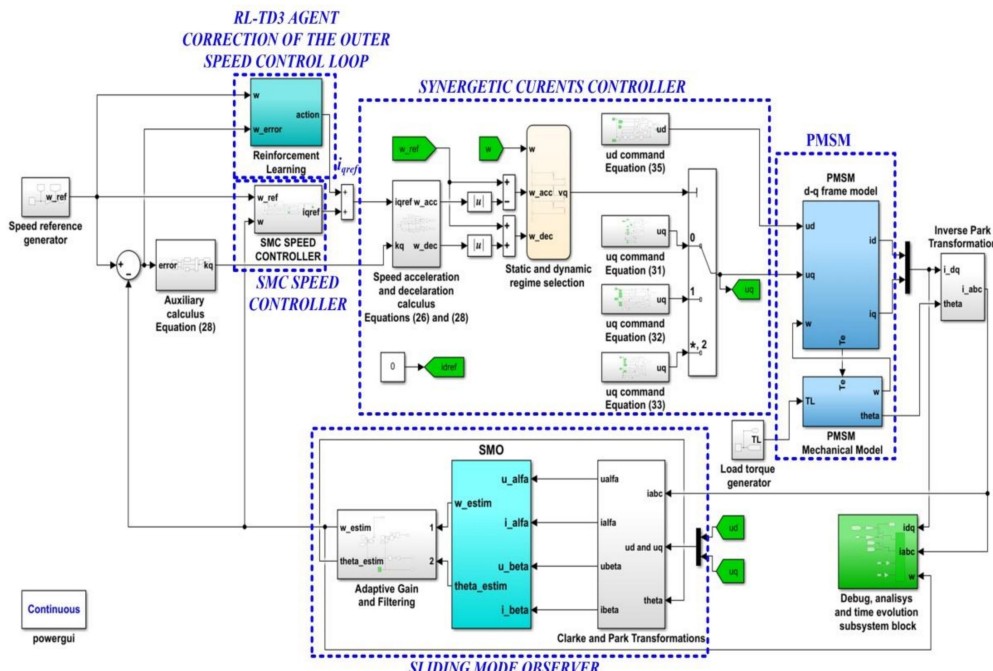

**Figure 22.** Block diagram of the MATLAB/Simulink implementation for the PMSM control based on SMC and synergetic controllers using the RL-TD3 agent for the correction of $i_{qref}$.

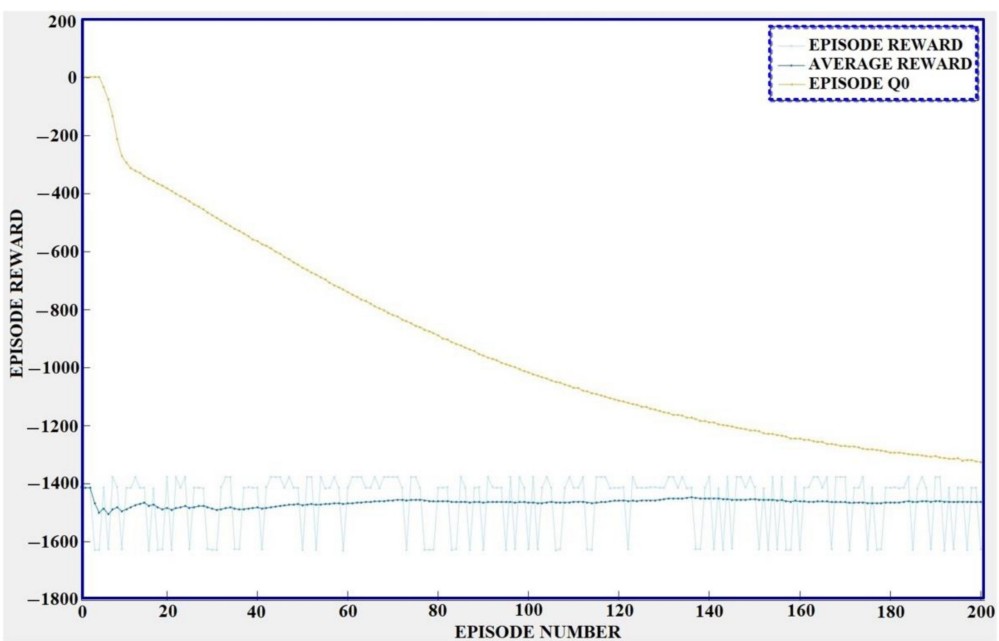

**Figure 23.** Training stage of the RL-TD3 agent for the correction of $i_{qref}$.

### 4.3. Reinforcement Learning Agent for the Correction of the Inner Currents Control Loop Using SMC and Synergetic Control

The block diagram of the implementation in MATLAB/Simulink for the PMSM control using SMC and synergetic controllers, resulting in improved performance using the RL-TD3 agent in the inner control loop, is presented in Figure 24. The correction signals of the RL-TD3 agent will be added to the control signals $u_{dref}$ and $u_{qref}$, the RL block structure is similar to that of Figure 9, and the Reward is given by Equation (11).

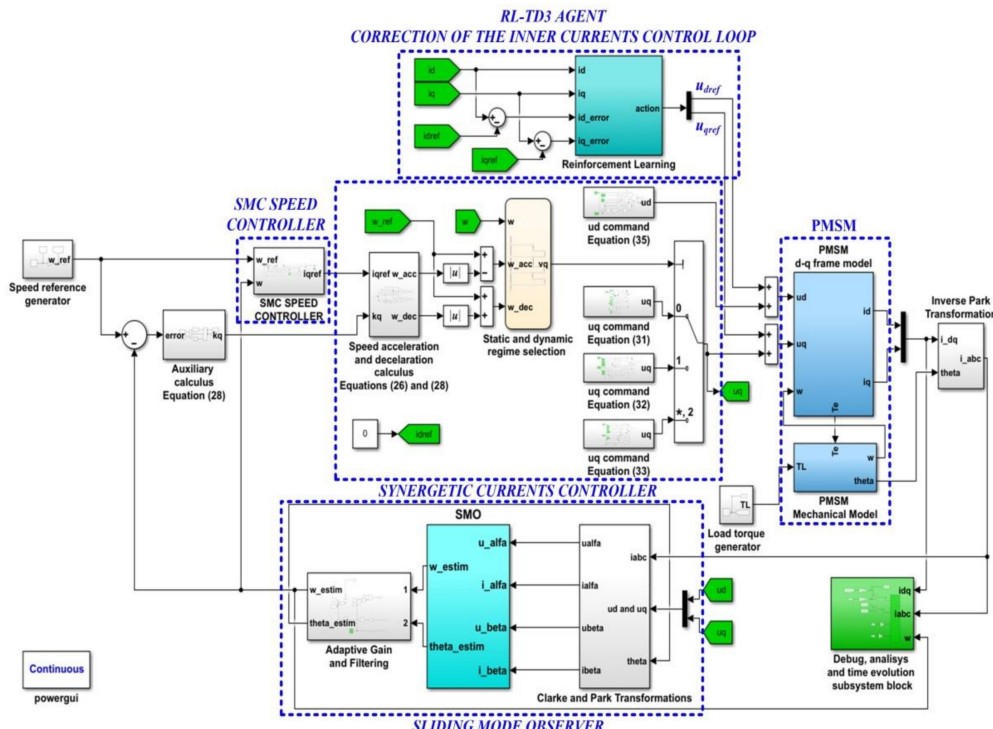

**Figure 24.** Block diagram of the MATLAB/Simulink implementation for the PMSM control based on SMC and synergetic controllers using the RL-TD3 agent for the correction of $u_{dref}$ and $u_{qref}$.

The training time for this case is 10 h, 29 min, and 14 s. The graphical results for the training stage of the RL-TD3 agent for the correction of $u_{dref}$ and $u_{qref}$ are shown in Figure 25.

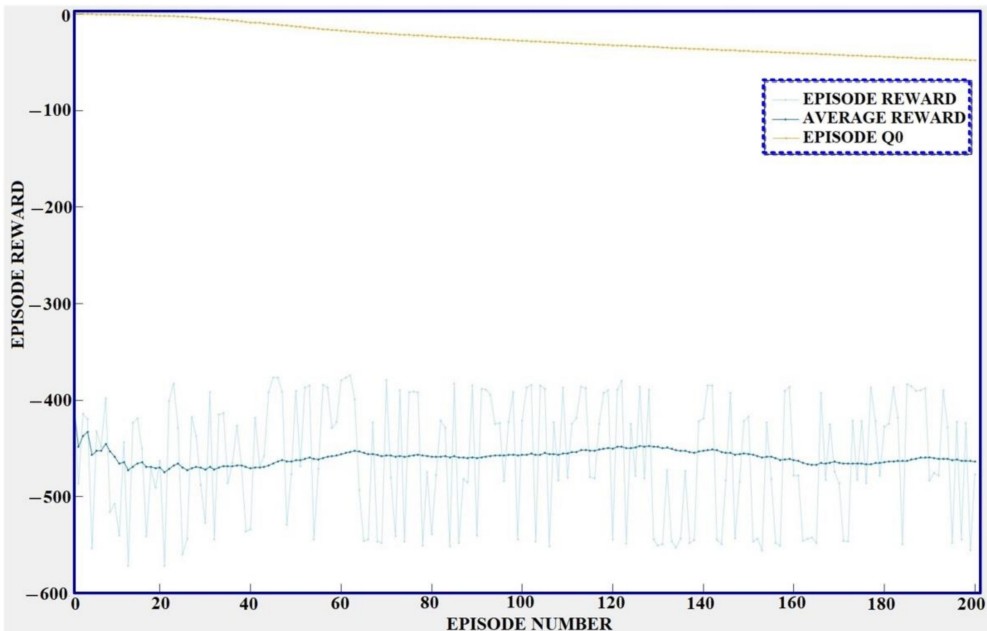

**Figure 25.** Training stage of the RL-TD3 agent for the correction of $u_{dref}$ and $u_{qref}$.

*4.4. Reinforcement Learning Agent for the Correction of the Outer Speed Control Loop and Inner Currents Control Loop Using SMC and Synergetic Control*

The block diagram of the implementation in MATLAB/Simulink for the PMSM control using SMC and synergetic controllers, resulting in the improved performance using the

RL-TD3 agent in the outer and inner control loops, is presented in Figure 26. The correction signals of the RL-TD3 agent will be added to the control signals $i_{qref}$, $u_{dref}$, and $u_{qref}$, the RL block structure is similar to that of Figure 12, and the Reward is given by Equation (12).

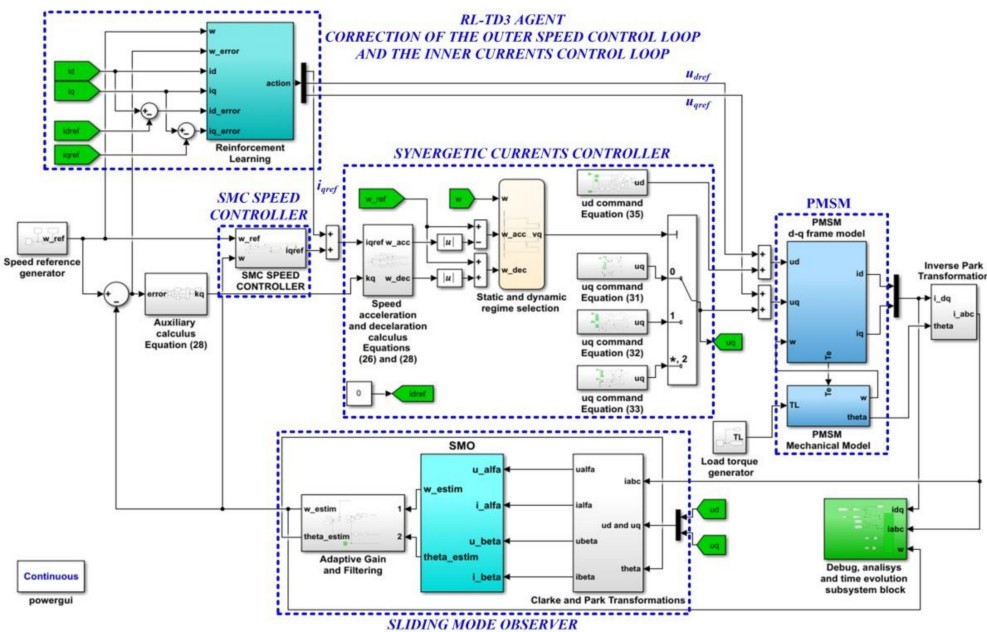

**Figure 26.** Block diagram of the MATLAB/Simulink implementation for the PMSM control based on the RL-TD3 agent for the correction of $u_{dref}$, $u_{qref}$, and $i_{qref}$.

The training time for this case is 11 h, 13 min, and 54 s. The graphical results for the training stage of the RL-TD3 agent for the correction of $i_{qref}$, $u_{dref}$, and $u_{qref}$ are shown in Figure 27.

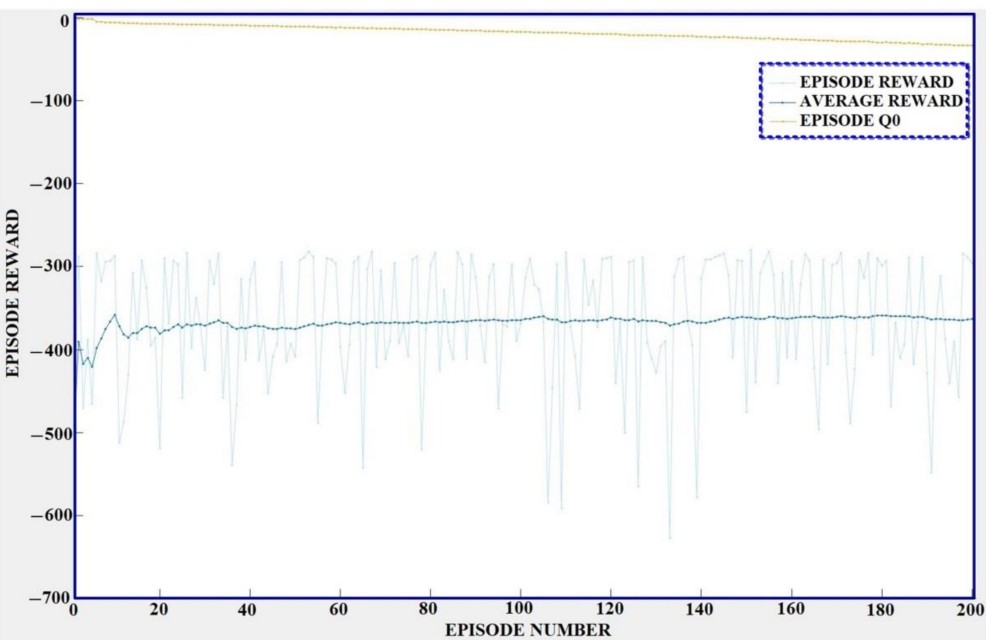

**Figure 27.** Training stage of the RL-TD3 agent for the correction of $u_{dref}$, $u_{qref}$, and $i_{qref}$.

## 5. Numerical Simulations

For a PMSM with the nominal parameters presented in Table 1, using MATLAB/ Simulink, Figure 28 shows the time evolution of the signals of interest of the PMSM control system. For the variation in the reference signal of the PMSM rotor speed from 800 to 1200 rpm at 0.5 s and a load torque of 0.5 Nm, a zero steady-state error and a response time of 30 ms are noted.

**Table 1.** PMSM nominal parameters.

| Parameter | Value | Unit |
|---|---|---|
| Stator resistance—$R_s$ | 2.875 | $\Omega$ |
| Inductances on *d-q* axis—$L_d$, $L_q$ | 0.0085 | H |
| Combined inertia of PMSM rotor and load—$J$ | $8 \times 10^3$ | kg·m$^2$ |
| Combined viscous friction of PMSM rotor and load—$B$ | 0.01 | N·m·s/rad |
| Flux induced by the permanent magnets of the PMSM rotor in the stator phases—$\lambda_0$ | 0.175 | Wb |
| Pole pairs number—$n_p$ | 4 | - |

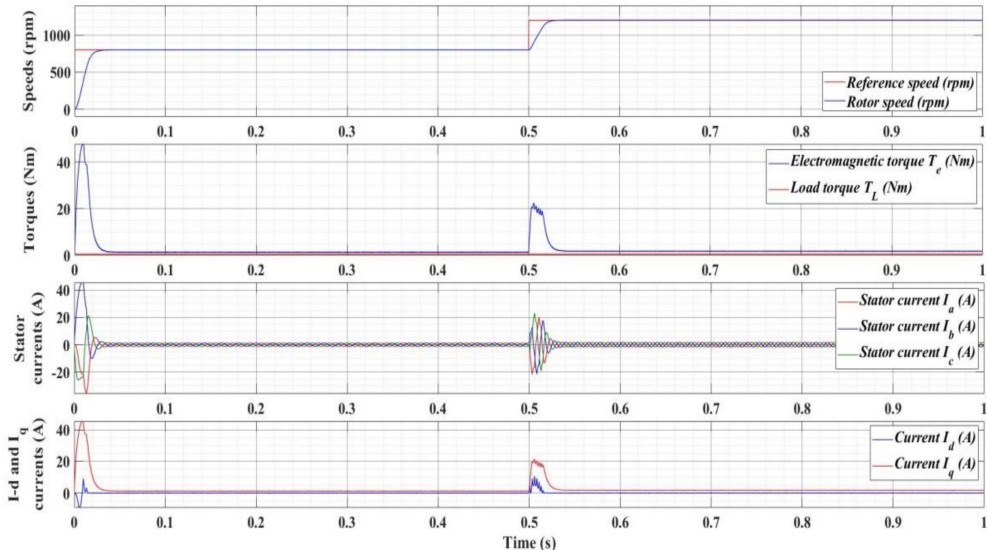

**Figure 28.** Time evolution for the numerical simulation of the PMSM control system based on the FOC-type strategy.

As a result of the creation of RL-TD3 agents, their training, and the numerical simulations related to the cases in Sections 3.1–3.3 of the Section 3, Figures 29–31 show the corresponding results obtained.

In the case of the RL agent for the correction of the outer loop for speed control (Figure 29), the stator currents remain identical to those in the classic case of the FOC-type control structure, but the response time improves very little, in the sense that it decreases by approximately 1 ms.

In the case of the RL-TD3 agent for the correction of the inner current control loop (Figure 30), an improvement is noted in the response of the control system, i.e., a shorter response time of 25 ms, but the stator currents at start-up increase by 50%. In this case, the current decreases to the usual values by decreasing the hysteresis of the current controllers, but a value of the response time of approximately 28 ms is obtained.

In the case of the RL-TD3 agent for the correction of the outer speed control loop and the inner current control loop (Figure 31), a very good global response is obtained, namely the response time decreases to 24 ms under the conditions where the stator currents remain similar to those in the classic case of the FOC-type control structure.

The efficiency of using the RL is thereby demonstrated by improving the performance of the PMSM control system when using the FOC-type strategy with PI-type controllers for speed control and hysteresis ON/OFF for current control. Depending on the application, among the presented variants, we choose the one that achieves an acceptable compromise between the decrease in the response time and the increase in the stator currents.

Based on articles [42,43], in which the FOC-type structure with SMC-type controllers was chosen for speed control and synergetic-type controllers were chosen for current control, and peak performance was obtained on benchmarks (including for the PMSM control), the following simulations validate that these performances can be further improved by using the RL.

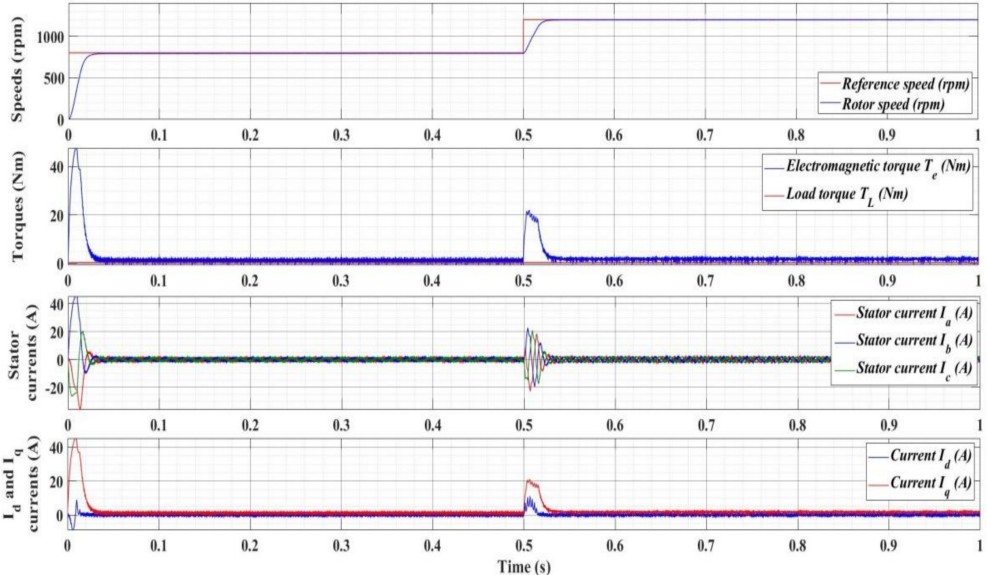

**Figure 29.** Time evolution for the numerical simulation of the PMSM control system based on the RL-TD3 agent for the correction of $i_{qref}$.

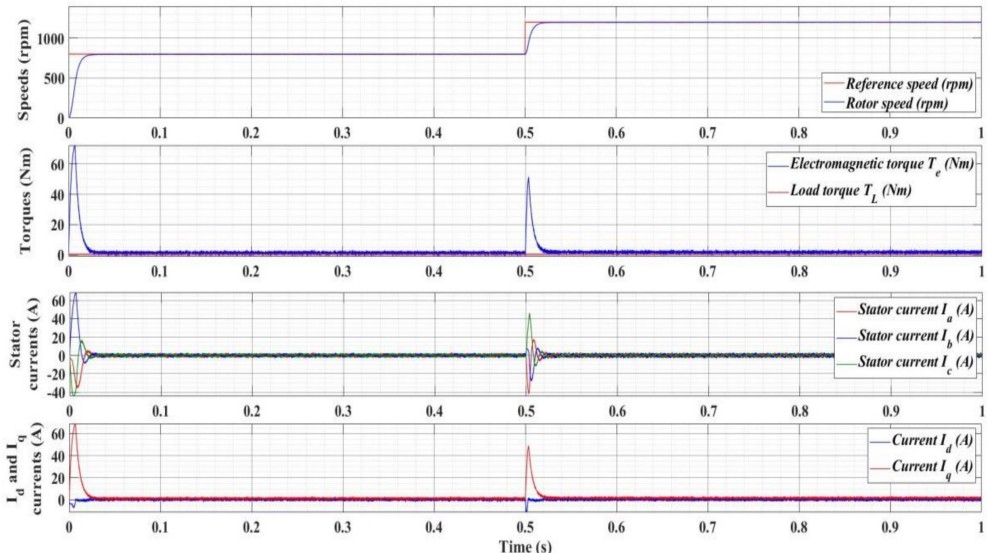

**Figure 30.** Time evolution for the numerical simulation of the PMSM control system based on the RL-TD3 agent for the correction of $u_{dref}$ and $u_{qref}$.

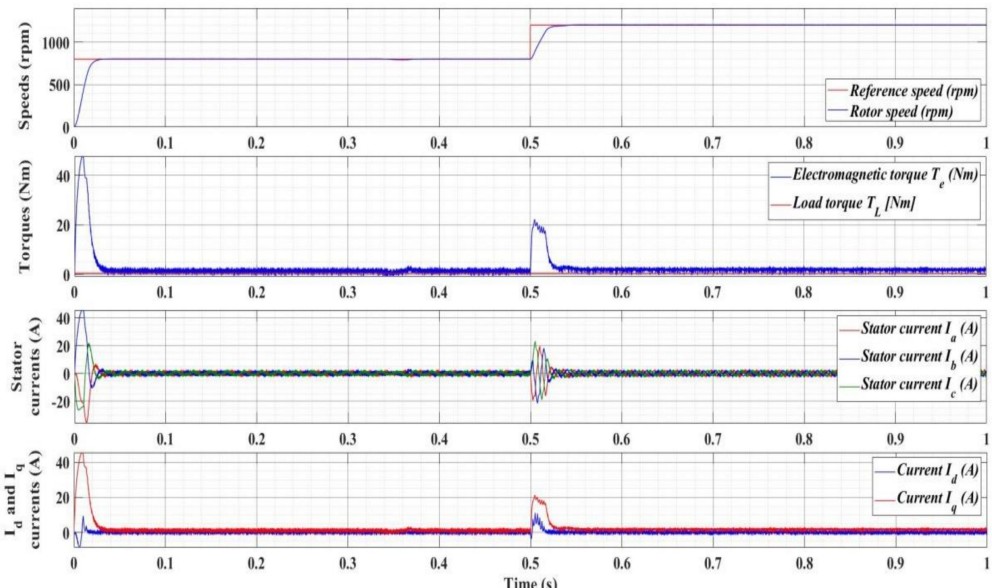

**Figure 31.** Time evolution for the numerical simulation of the PMSM control system based on the RL-TD3 agent for the correction of $u_{dref}$, $u_{qref}$, and $i_{qref}$.

Thus, in Figure 32, for a FOC-type structure based on the SMC and synergetic controllers, the response of the PMSM control system is presented under the same conditions in which the FOC-type structure is based on PI and hysteresis ON/OFF controllers. Very good control performance is noted, with a response time of 14.1 ms.

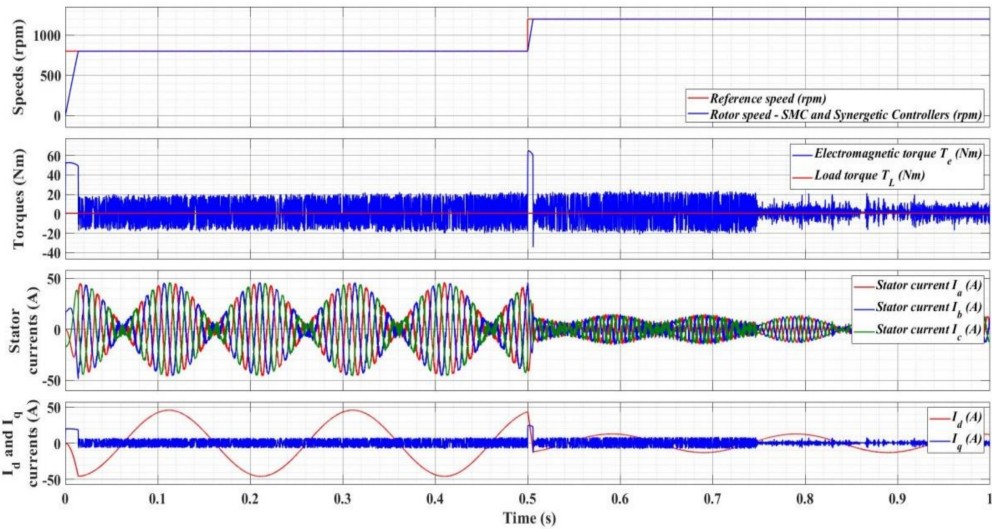

**Figure 32.** Time evolution for the numerical simulation of the PMSM control system based on control using SMC and synergetic controllers.

Moreover, Figures 33–35 show the response of the PMSM control system under the same conditions as those presented above, in which the control signal is supplemented with the output of an RL-TD3 agent for the speed control loop, for the current control loop, and for both control loops.

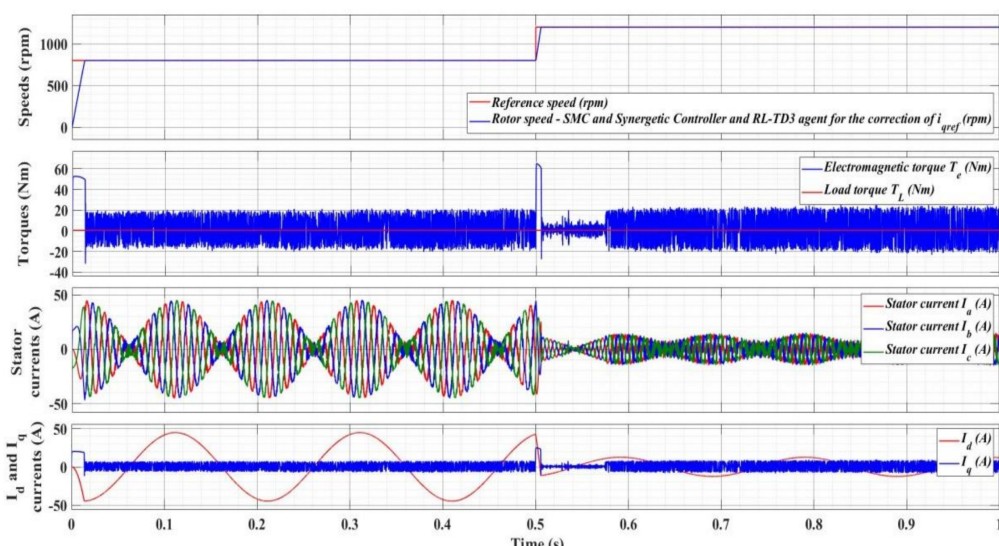

**Figure 33.** Time evolution for the numerical simulation of the PMSM control system based on control using SMC and synergetic controllers using an RL-TD3 agent for the correction of $i_{qref}$.

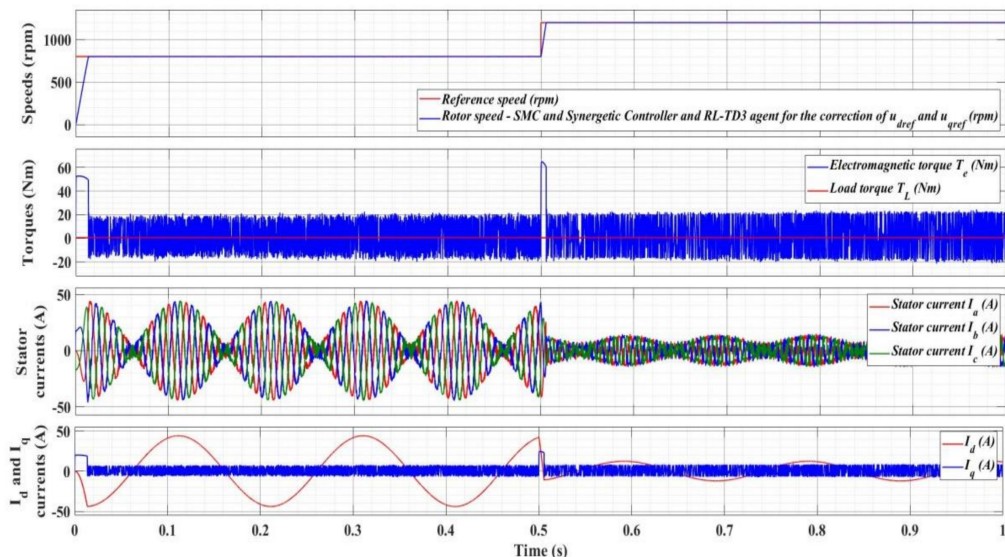

**Figure 34.** Time evolution for the numerical simulation of the PMSM control system based on control using SMC and synergetic controllers using an RL-TD3 agent for the correction of $u_{dref}$ and $u_{qref}$.

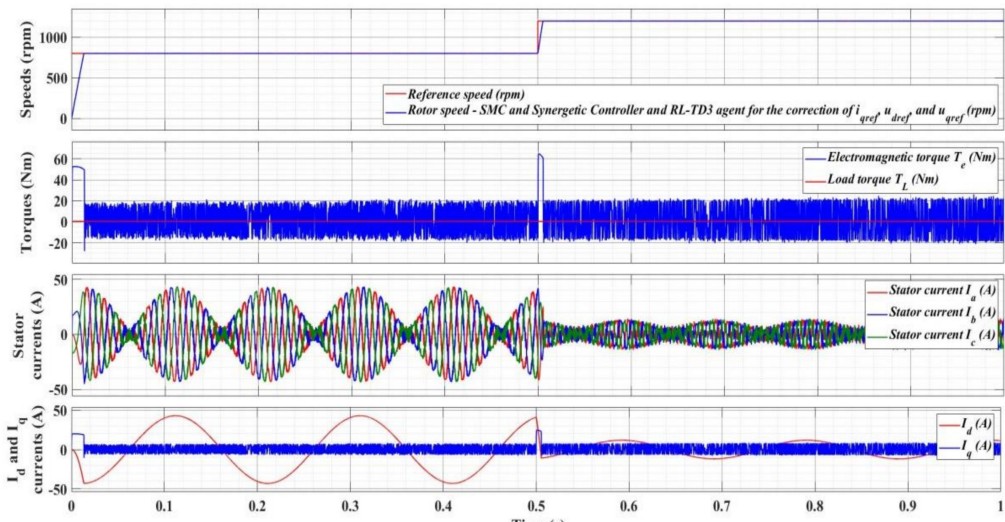

**Figure 35.** Time evolution for the numerical simulation of the PMSM control system based on control using SMC and synergetic controllers using an RL-TD3 agent for the correction of $u_{dref}$, $u_{qref}$, and $i_{qref}$.

The validation of the robustness of the PMSM control system based on SMC and synergetic controllers, and the three variants of control signal correction implemented by means of the RL-TD3 agent, were achieved by modifying the load torque $T_L$ and the combined inertia of the PMSM rotor and load $J$, which are considered to be disturbances. Thus, numerical simulations were performed for a value of $T_L = 2$ Nm, to which a 0.2 Nm magnitude uniformly distributed noise and a 50% $J$ parameter increase were added. Following the numerical simulations, Figures 36–39 shows the response of the PMSM control system under parametric changes. It can be noted that the PMSM control system retains its previous performance, noting that the response time increases by about 0.4 ms for each variant of the control system.

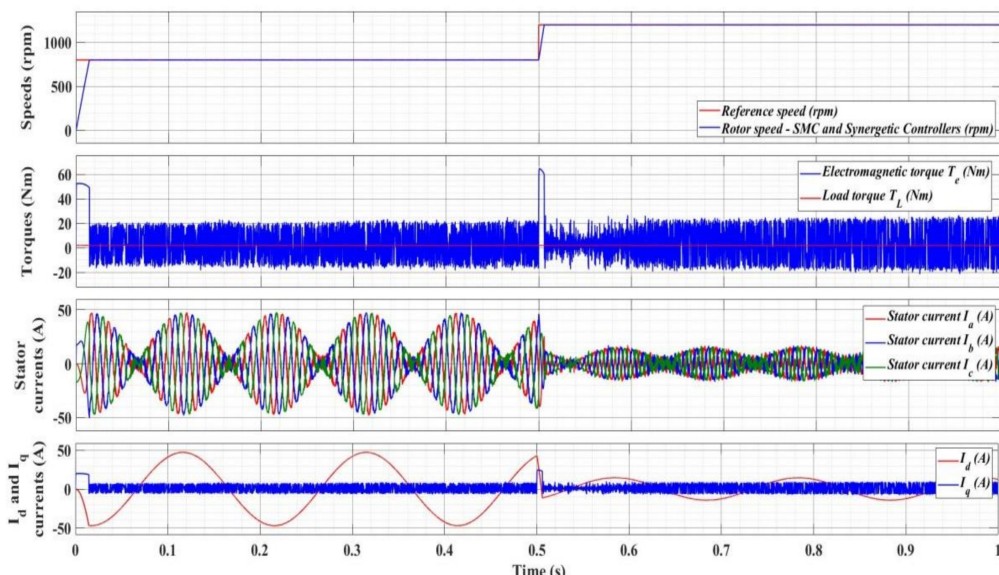

**Figure 36.** Time evolution for the numerical simulation of the PMSM control system based on control using SMC and synergetic controllers: $T_L = 2$ Nm with 0.2 Nm magnitude of uniformly distributed noise and 50% increase in $J$ parameter.

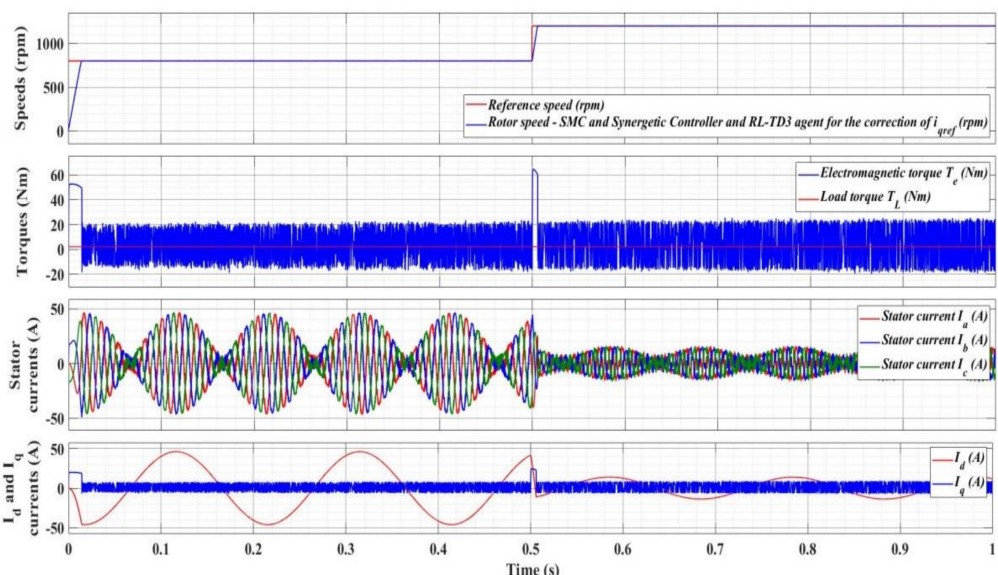

**Figure 37.** Time evolution for the numerical simulation of the PMSM control system based on control using SMC and synergetic controllers using an RL-TD3 agent for the correction of $i_{qref}$: $T_L$ = 2 Nm with 0.2 Nm magnitude of uniformly distributed noise and 50% increase in $J$ parameter.

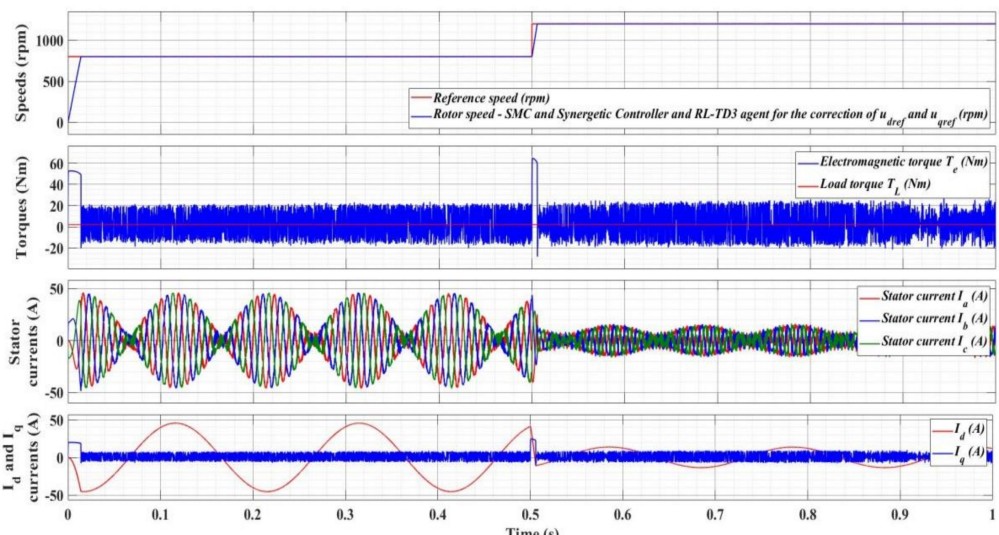

**Figure 38.** Time evolution for the numerical simulation of the PMSM control system based on control using SMC and synergetic controllers using an RL-TD3 agent for the correction of $u_{dref}$ and $u_{qref}$: $T_L$ = 2 Nm with 0.2 Nm magnitude of uniformly distributed noise and 50% increase in $J$ parameter.

Figure 40 shows the comparative response of the PMSM control system for an 800 rpm reference speed step, a load torque of 0.5 Nm for the of PMSM control systems based on SMC and synergetic controllers, and three variants of this type of control system using an RL-TD3 agent for correction of the command signals (outer and inner control loop).

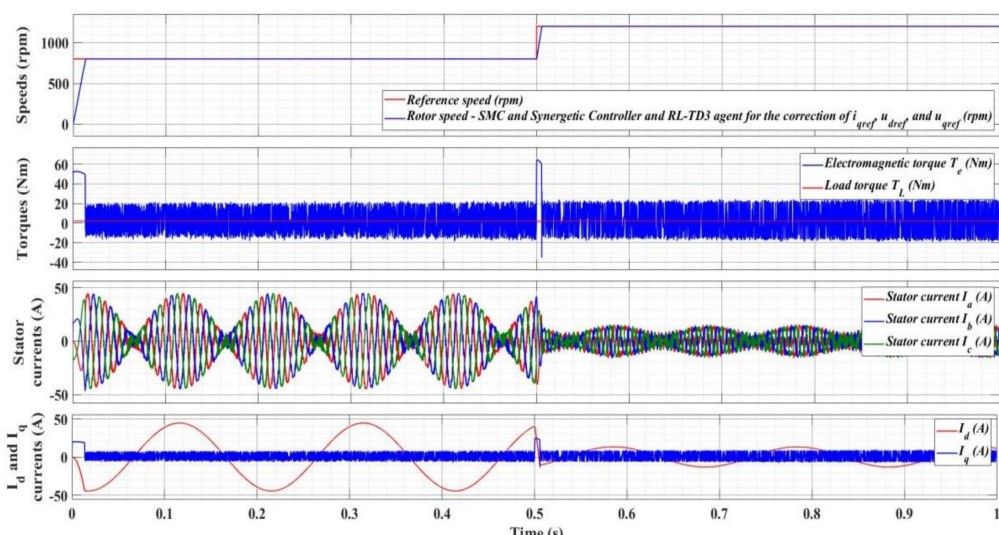

**Figure 39.** Time evolution for the numerical simulation of the PMSM control system based on control using SMC and synergetic controllers using an RL-TD3 agent for the correction of $u_{dref}$, $u_{qref}$, and $i_{qref}$: $T_L$ = 2 Nm with 0.2 Nm magnitude of uniformly distributed noise and 50% increase in $J$ parameter.

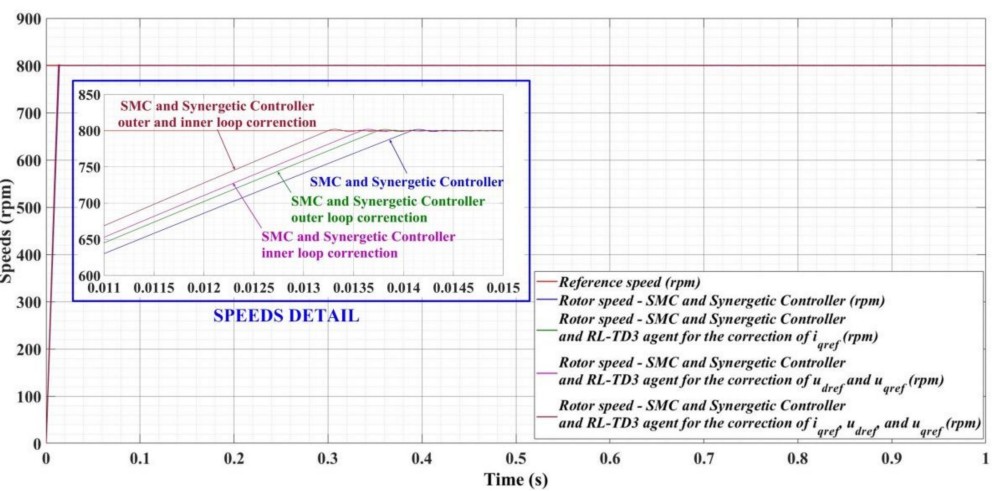

**Figure 40.** Comparison of the speed time evolution for the numerical simulation of the PMSM control system based on SMC and synergetic controllers using an RL-TD3 agent for the outer and inner loop correction.

Table 2 shows the comparative performance of these types of PMSM control system variants, i.e., the response time and the ripple of the PMSM rotor speed error signal given by Equation (48). In all these cases the overshooting is zero, and the steady-state error is less than 0.1%.

$$\omega_{rip} = \sqrt{\frac{1}{N}\sum_{i=1}^{N}\left(\omega(i) - \omega_{ref}(i)\right)^2} \tag{48}$$

where: $N$ represents the number of samples, $\omega$ represents the PMSM rotor speed, and $\omega_{ref}$ represents the reference speed.

**Table 2.** PMSM control system based on SMC and synergetic controllers using RL-TD3 agent performances.

| Controllers | Response Time [ms] | Speed Ripple [rpm] |
|---|---|---|
| SMC and synergetic | 14.1 | 54.78 |
| SMC and synergetic using RL-TD3 agent for the correction of $i_{qref}$. | 13.7 | 54.15 |
| SMC and synergetic using RL-TD3 agent for the correction of $u_{dref}$ and $u_{qref}$ | 13.5 | 53.83 |
| SMC and synergetic using RL-TD3 agent for the correction of $u_{dref}$, $u_{qref}$, and $i_{qref}$ | 13.2 | 53.18 |

A previous study [43] described in detail the implementation of real-time control algorithms (such as SMC and synergetic controllers) in an S32K144 development kit containing an S32K144 evaluation board (S32K144EVB-Q100), a DEVKIT-MOTORGD board based on a SMARTMOS GD3000 pre-driver, and a Linix 45ZWN24-40 PMSM type. The controller of the development platform was an S32K144 MCUm, which is a 32-bit Cortex M4F type having a time base of 112 MHz with 512 KB of flash memory and 54 KB of RAM. The presentation in real time of the improvement resulting from the current article in the PMSM control system on the same benchmark was omitted, because it was considered less important than the presentation of the RL-TD3 agent control structures and the advantages they provided.

## 6. Conclusions

This paper presents the FOC-type control structure of a PMSM, which is improved in terms of performance by using a RL technique. Thus, the comparative results are presented for the case where the RL-TD3 agent is properly trained and provides correction signals that are added to the control signals $u_d$, $u_q$, and $i_{qref}$. The FOC-type control structure for the PMSM control based on an SMC speed controller and synergetic current controller is also presented. To improve the performance of the PMSM control system without using controllers having a more complicated mathematical description, the advantages provided by the RL on process control can also be used. This improvement is obtained using the correction signals provided by a trained RL-TD3 agent, which is added to the control signals $u_d$, $u_q$, and $i_{qref}$. A speed observer is also implemented for estimating the PMSM rotor speed. The parametric robustness of the proposed PMSM control system is proved by very good control performances achieved even when the uniformly distributed noise is added to the load torque $T_L$, and under high variations in the load torque $T_L$ and the moment of inertia $J$. Numerical simulations are used to prove the superiority of the control system that uses the RL-TD3 agent.

**Author Contributions:** Conceptualization, M.N. and C.-I.N.; Data curation, M.N., C.-I.N. and D.S.; Formal analysis, M.N., C.-I.N. and D.S.; Funding acquisition, M.N. and D.S.; Investigation, M.N., C.-I.N. and D.S.; Methodology, M.N., C.-I.N. and D.S.; Project administration, M.N. and D.S.; Resources, M.N. and D.S.; Software, M.N. and C.-I.N.; Supervision, M.N. and D.S.; Validation, M.N. and D.S.; Visualization, M.N., C.-I.N. and D.S.; Writing—original draft, M.N., C.-I.N. and D.S.; Writing—review and editing, M.N., C.-I.N. and D.S. All authors have read and agreed to the published version of the manuscript.

**Funding:** This work was partially supported by European Regional Development Fund Competitiveness Operational Program, project TISIPRO, ID: P_40_416/105736, 2016–2021 and with funds from the Ministry of Research and Innovation—Romania as part of the NUCLEU Program: PN 19 38 01 03.

**Institutional Review Board Statement:** Not applicable.

**Informed Consent Statement:** Not applicable.

**Data Availability Statement:** Not applicable.

**Conflicts of Interest:** The authors declare no conflict of interest.

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
