# Peer review of "Improvement of PMSM Sensorless Control Based on Synergetic and Sliding Mode Controllers Using a Reinforcement Learning Deep Deterministic Policy Gradient Agent†"

_energies, doi:10.3390/en15062208_

Round 1

Reviewer 1 Report

This paper is well written and sounds interesting. I would like to suggest acceptance after the authors addressed the following comments.

  1. Please discuss the motivation of this study in detail.
  2. I suggest the authors consider the comparison with the existing designs such as other ML algorithms.

Author Response

Dear reviewer, thanks for your recommendations.

  1. The intelligent control system has a special role in the development and improvement of control systems. Among them we mention the RL agents, which are characterized by the fact that they do not use the mathematical description of the controlled system, but use input signals which contain information on the state of the system and provide control actions to optimize a reward which consists of signals containing information on the controlled process [32–37]. Moreover, starting from 4 papers by the authors based on the RL agent, which demonstrates the capacity of this type of algorithm to improve the performance of the linear control algorithms of the PMSM (PI type algorithm [26,38], LQR control algorithm [39], Feedback Linearization control algorithm [40]), this article demonstrates that a cascade control structure containing nonlinear SMC and synergetic control algorithms can also exhibit improved performance by using an RL agent. This paper is a follow-up on paper [38] presented at the 21st International Symposium on Power Electronics (Ee 2021), which shows an improvement of the performance of the FOC-type control structure of the PMSM by using an RL agent. The type of RL used is the Twin-Delayed Deep Deterministic Policy Gradient (TD3) agent which is an extension and improved variant of the DDPG agent [41]. Moreover, based on the fact that the FOC-type control structure where the controller of the outer speed control loop is of SMC type and the controllers of the inner current control loop are of synergetic type provides peak performance of the control system [42,43], this paper also presents the improvement of the performance of this control structure by using an RL agent.
  2. The commonly used Machine Learning (ML) techniques are the follows: Linear Regression, Decision Tree, Support Vector Machine, Neural Networks and Deep Learning, and Ensemble of Tree. Generally, the three main types of machine learning strategies are the follows: unsupervised learning, supervised learning, and RL. First, unsupervised learning is typically utilized in the fields of data clustering and dimensionality reduction. Supervised learning deals mainly with classification and regression problems. RL is a framework for learning the relationship between states and actions. Ultimately, the agent can maximize the expected reward and learn how to complete a mission. This is a very strong analogy with the control of an industrial process. While the other types of ML can be used to estimate certain parameters, RL is recommended for the control of an industrial process. We can tell that RL is an especially powerful form of artificial intelligence, and we’re sure to see more progress from these teams [*]. Moreover, the Twin-Delayed Deep Deterministic Policy Gradient Agent - RL-TD3 (used in this article) is an improved version of the Deep Deterministic Policy Gradient Agents - DDPG and is considered the most suitable RL agent for industrial process control [**]. In addition, the authors' articles [26,38,39,40] present comparatively the results of control systems improved by using the RL-TD3 agent or optimized control law parameters by means of computational intelligence algorithms: Particle Swarm Optimization (PSO) algorithm, Simulated Annealing (SA) algorithm, Genetic Algorithm (GA), and Gray Wolf Optimization (GWO) algorithm. The conclusion clearly shows the superiority of the performance of the PMSM control system in case of using the RL-TD3 agent. However, we consider that the control structures and the comparative results presented are relevant for this article, but the explicit requirement to make a comparison with other ML algorithms clearly entails writing another paper.

[*]https://towardsdatascience.com/10-machine-learning-methods-that-every-data-scientist-should-know-3cc96e0eeee9

[**]MathWorks—Reinforcement Learning Toolbox™ User's Guide. Available online: https://www.mathworks.com/help/reinforcement-learning/getting-started-with-reinforcement-learning-toolbox.html?s_tid=CRUX_lftnav (accessed on 4 November 2021).

[26] Nicola, M.; Nicola, C.-I. Tuning of PI Speed Controller for PMSM Control System Using Computational Intelligence. In Proceedings of the 21st International Symposium on Power Electronics (Ee), Novi Sad Serbia, 27-30 October 2021, pp. 1-6.

[38] Nicola, M.; Nicola, C.-I. Improvement of PMSM Control Using Reinforcement Learning Deep Deterministic Policy Gradient Agent. In Proceedings of 21st the International Symposium on Power Electronics (Ee), Novi Sad, Serbia, 27-30 October 2021; pp. 1–6.

[39] Nicola, M.; Nicola, C.-I. Improved Performance for PMSM Control System Based on LQR Controller and Computational Intelligence. In Proceedings of the International Conference on Electrical, Computer and Energy Technologies (ICECET), Cape Town, South Africa, 9-10 December 2021; pp. 1–6.

[40] Nicola, M.; Nicola, C.-I. Improved Performance for PMSM Control System Based on Feedback Linearization and Computational Intelligence. In Proceedings of the International Conference on Electrical, Computer and Energy Technologies (ICECET), Cape Town, South Africa, 9-10 December 2021; pp. 1–6.

Reviewer 2 Report

  1. It is a valuable work, because PMSM sensorless control has important significance.
  2. The proposed RL-DDPG lacks sufficient comparative verification with popular controller, especially lacks experimental results. From results in Figure 30 to Figure 35, the system dynamic response under parameters changing shows less improvement, because response time is long and overmodulation is big.
  3. The paper writing should avoid the basic process of simulation, including the well-known concepts and simulink model, such as Figures 3, 5, 6, 8.

Author Response

Dear reviewer, thanks for your recommendations.

  • 4 papers by the authors based on the RL agent demonstrate the capacity of this type of algorithm to improve the performance of the linear control algorithms of the PMSM (PI type algorithm [26,38], LQR control algorithm [39], Feedback Linearization control algorithm [40]), this article demonstrates that a cascade control structure containing nonlinear SMC and synergetic control algorithms can also exhibit improved performance by using an RL agent. RL is a framework for learning the relationship between states and actions. Ultimately, the agent can maximize the expected reward and learn how to complete a mission. This is a very strong analogy with the control of an industrial process. While the other types of ML can be used to estimate certain parameters, RL is recommended for the control of an industrial process. In addition, the authors' articles [26,38,39,40] present comparatively the results of control systems improved by using the RL-TD3 agent or optimized control law parameters by means of computational intelligence algorithms: Particle Swarm Optimization (PSO) algorithm, Simulated Annealing (SA) algorithm, Genetic Algorithm (GA), and Gray Wolf Optimization (GWO) algorithm. The conclusion clearly shows the superiority of the performance of the PMSM control system in case of using the RL-TD3 agent. Furthermore, this article presents the improvement to the PMSM control system by also using the agent RL-TD3 in the case of non-linear SMC and synergetic control algorithms. Moreover, the cascade control structure based on SMC (outer speed control loop) and synergetic (inner current control loop) control algorithms for the control of a PMSM was proposed by the authors in [43] and peak performance was obtained for a benchmark PMSM. Furthermore, [43] describes in detail the implementation of real-time control algorithms in an S32K144 development kit which contains an S32K144 evaluation board (S32K144EVB-Q100), DEVKIT-MOTORGD board based on SMARTMOS GD3000 pre-driver and Linix 45ZWN24-40 PMSM type. The controller of the development platform is S32K144 MCU which is of 32 bit Cortex M4F type, which has a time base of 112 MHz with 512 KB of flash memory and 54 KB of RAM. The presentation in real time of the improvement brought in the current article to the PMSM control system on the same benchmark was omitted, being considered less important than the presentation of the RL-TD3 agent control structures and the advantages brought by them. The improvement of the performance is about 10% see Table 2.

Table 2. PMSM control system based on SMC and synergetic using RL-TD3 agent performances

Controllers

Response time [ms]

Speed ripple [rpm]

SMC and synergetic

14.1

54.78

SMC and synergetic using RL-TD3 agent for the correction of iqref.

13.7

54.15

SMC and synergetic using RL-TD3 agent for the correction of udref and uqref

13.5

53.83

SMC and synergetic using RL-TD3 agent for the correction of udref, uqref, and iqref

13.2

53.18

  • Your observation, in the opinion of the authors (who in turn are also reviewers at MDPI) is subjective. In this regard, with all due respect, we quote some elements by another reviewer of this article regarding the aspects indicated by you. “This manuscript deal with design reinforcement learning-based deep deterministic policy gradient method for the field-oriented approach of the permanent magnet synchronous motor control. The Introduction section fully describes the approaches in RL agent development. The second section is shown the RL design with the problem statement and how to train the agents. The third section focuses on integrating RL agents into the FOC-control approach.  The structural schemes and flowcharts seem adequate to me. The Authors have shown a lot of figures with the training sets and it could be useful to young researchers. The fourth section deal with the PMSM with agents' control system synthesis. The system is fully described, the structural scheme was shown and the mathematical descriptions are demonstrated. The authors described synthesis in detail and it could be useful to other researchers in replicating this experience. The Simulation section shows the time response PMSM with the development method and the summary table is shown the correctness of the development methods. In the Conclusion, the Authors was summarising the results correctly. In my opinion, this paper could be extremely useful for young researchers in the area control system with the agents' approach”.  

[26] Nicola, M.; Nicola, C.-I. Tuning of PI Speed Controller for PMSM Control System Using Computational Intelligence. In Proceedings of the 21st International Symposium on Power Electronics (Ee), Novi Sad Serbia, 27-30 October 2021, pp. 1-6.

[38] Nicola, M.; Nicola, C.-I. Improvement of PMSM Control Using Reinforcement Learning Deep Deterministic Policy Gradient Agent. In Proceedings of 21st the International Symposium on Power Electronics (Ee), Novi Sad, Serbia, 27-30 October 2021; pp. 1–6.

[39] Nicola, M.; Nicola, C.-I. Improved Performance for PMSM Control System Based on LQR Controller and Computational Intelligence. In Proceedings of the International Conference on Electrical, Computer and Energy Technologies (ICECET), Cape Town, South Africa, 9-10 December 2021; pp. 1–6.

[40] Nicola, M.; Nicola, C.-I. Improved Performance for PMSM Control System Based on Feedback Linearization and Computational Intelligence. In Proceedings of the International Conference on Electrical, Computer and Energy Technologies (ICECET), Cape Town, South Africa, 9-10 December 2021; pp. 1–6.

[43] Nicola, M.; Nicola, C.-I. Sensorless Fractional Order Control of PMSM Based on Synergetic and Sliding Mode Controllers. Electronics 2020, 9, 1494.

Reviewer 3 Report

This manuscript deal with design reinforcement learning-based deep deterministic policy gradient method for the field-oriented approach of the permanent magnet synchronous motor control. The Introduction section fully describes the approaches in RL agent development. The second section is shown the RL design with the problem statement and how to train the agents. The third section focuses on integrating RL agents into the FOC-control approach.  The structural schemes and flowcharts seem adequate to me. The Authors have shown a lot of figures with the training sets and it could be useful to young researchers. The fourth section deal with the PMSM with agents' control system synthesis. The system is fully described, the structural scheme was shown and the mathematical descriptions are demonstrated. The authors described synthesis in detail and it could be useful to other researchers in replicating this experience. The Simulation section shows the time response PMSM with the development method and the summary table is shown the correctness of the development methods. In the Conclusion, the Authors was summarising the results correctly. 

In my opinion, this paper could be extremely useful for young researchers in the area control system with the agents' approach. 

Author Response

Dear reviewer, thanks for your recommendations.

Round 2

Reviewer 2 Report

The issues raised before have been revised.